# A switch in transcription and cell fate governs the onset of an epigenetically-deregulated tumor in *Drosophila*

**Joana Torres[1], Remo Monti[1], Ariane L Moore[1,2], Makiko Seimiya[1], Yanrui Jiang[1], Niko Beerenwinkel[1,2], Christian Beisel[1], Jorge V Beira[1†*], Renato Paro[1,3†*]**

[1]Department of Biosystems Science and Engineering, ETH Zürich, Basel, Switzerland; [2]Swiss Institute of Bioinformatics, Basel, Switzerland; [3]Faculty of Science, University of Basel, Basel, Switzerland

**Abstract** Tumor initiation is often linked to a loss of cellular identity. Transcriptional programs determining cellular identity are preserved by epigenetically-acting chromatin factors. Although such regulators are among the most frequently mutated genes in cancer, it is not well understood how an abnormal epigenetic condition contributes to tumor onset. In this work, we investigated the gene signature of tumors caused by disruption of the *Drosophila* epigenetic regulator, *polyhomeotic (ph).* In larval tissue *ph* mutant cells show a shift towards an embryonic-like signature. Using loss- and gain-of-function experiments we uncovered the embryonic transcription factor *knirps (kni)* as a new oncogene. The oncogenic potential of *kni* lies in its ability to activate JAK/STAT signaling and block differentiation. Conversely, tumor growth in *ph* mutant cells can be substantially reduced by overexpressing a differentiation factor. This demonstrates that epigenetically derailed tumor conditions can be reversed when targeting key players in the transcriptional network.

DOI: https://doi.org/10.7554/eLife.32697.001

**\*For correspondence:**
jbeira@gmail.com (JVB);
renato.paro@bsse.ethz.ch (RP)

†These authors contributed equally to this work

**Competing interests:** The authors declare that no competing interests exist.

## Introduction

During development, epigenetic regulators are responsible for controlling and restraining cellular plasticity. This tight regulation allows cells to differentiate faithfully and heritably towards a specific fate (*Roy and Hebrok, 2015*; *Wainwright and Scaffidi, 2017*). An appropriate balance between proliferation and differentiation is fundamental and multiple regulatory layers of transcription factors and epigenetic regulators are employed to accomplish the underlying transcriptional control (*Gonda and Ramsay, 2015*; *Piunti and Shilatifard, 2016*).

Many epigenetic regulators are evolutionary conserved and among the most commonly mutated genes in human cancer (*Piunti and Shilatifard, 2016*). Disruption of epigenetic constraints leads to global reorganization of the epigenome and changes in transcriptional profiles, which might provide a cellular state permissive for tumorigenesis (*Wainwright and Scaffidi, 2017*). Indeed, disturbed transcriptional profiles and putative oncogenic transcriptional regulators have recently gained significance as better alternatives for therapeutic targets in comparison to signaling pathways. Transcription factors (TFs) are less prone to be bypassed by alternative mutational events, and their perturbation can affect several cancer hallmarks. In addition, due to the complexity of transcriptional networks, it is unlikely that one TF functioning as an oncogenic driver can be entirely replaced by another (*Gonda and Ramsay, 2015*). For these reasons it is fundamental to identify the core transcriptional networks defining cancer cell types, and target those regulators crucial for survival (*Bonifer and Cockerill, 2015*).

**eLife digest** When an animal is developing as an embryo, different cells start to specialize into the specific cell types needed to form the tissues and organs of the body. How an individual cell commits to become a certain type of cell is mostly determined by which of the genes in its DNA are active. In animal cells, DNA is wrapped around proteins called histones, and one way that cells can maintain their distinct pattern of gene activity is via chemical tags on the histones. These tags can switch nearby genes on or off, and are added or removed by other proteins called epigenetic regulators. The epigenetic tags are also stably inherited when the cell divides, meaning that a cell's identity can be maintained over many cell generations.

If epigenetic regulators fail to work properly or get disrupted, the pattern of gene activity in a cell becomes altered. As a consequence, that cell can lose its identity and will often turn into a cancer cell. In fact, mutations in epigenetic regulators are found in several human cancers. It is not yet understood how these changes in gene expression lead cells to become cancerous.

Torres et al. have now analyzed an epigenetic regulator called Polyhomeotic in developing larvae of the fruit fly, *Drosophila melanogaster*. The results show that when Polyhomeotic is not produced the fly larvae develop tumors. Moreover, the mutant cells without Polyhomeotic had different gene expression profiles compared to normal cells. This in turn caused the mutant cells, which had previously committed to a certain fate, to become more like the unspecialized cells found in early embryos.

Torres et al. next showed that, among the genes that were incorrectly regulated when Polyhomeotic's activity was compromised, one gene called *knirps* was switched on by mistake, which led the mutant cells to become tumor cells. When the activity of *knirps* was reduced instead, almost no tumors grew. Additionally, Torres et al. found that the protein encoded by *knirps* activates a signaling pathway that keeps tumor cells unspecialized by blocking their normal progress to a more mature and specialized state – a process known as differentiation. Experimentally raising the levels of a different molecule that ultimately promotes differentiation caused the tumor cells to grow less.

These findings suggest that tumors caused when epigenetic regulation goes awry may be reversed by targeting key genes such as *knirps*. Further work is now needed to test whether these findings will also extend to humans. Forcing cancer cells from a highly dividing, non-specialized state into a dead-end, mature state may lead to new ways to treat cancer.

DOI: https://doi.org/10.7554/eLife.32697.002

Epigenetic regulators involved in preserving cellular identity are composed of two classes of chromatin proteins, the Polycomb (PcG) and the Trithorax group (TrxG), whose complementary functions maintain the repressed and active gene expression state, respectively (*Geisler and Paro, 2015*). PcG proteins are organized into two basic complexes, Polycomb repressive complex 1 and 2 (PRC1 and PRC2) (*Piunti and Shilatifard, 2016*). One example of classical PcG targets are homeotic genes encoding Homeobox TFs, first identified in *Drosophila* and responsible for correct spatial body development in flies (*Shah and Sukumar, 2010*; *Abate-Shen, 2002*). The altered expression of Hox genes in human tumors suggests important roles for both oncogenesis and tumor suppression (*Shah and Sukumar, 2010*), which further hints towards an role for PcG proteins in oncogenesis.

The tumor suppressive role of PcG proteins, in particular PRC1 members in *Drosophila* imaginal discs, has been extensively investigated (*Classen et al., 2009*; *Martinez et al., 2009*). However, the effects on the transcriptional landscape after PRC1 deregulation in tumorigenesis has only recently started to be assessed (*Bunker et al., 2015*; *Loubière et al., 2016*). Here, we show that loss of Polyhomeotic (*ph*), a member of the PRC1 complex, in eye-antennal imaginal discs of larvae leads to a reprogramming of cellular identity towards an embryonic state and a concomitant loss of differentiation markers. Among the reactivated genes is *knirps*, an orphan nuclear hormone receptor. Depletion of *knirps* revealed its vital role in tumor maintenance, while misexpression showed its capacity to drive tumorigenesis in otherwise wild-type tissues. Tumors initiated by *ph* disruption or misexpression of *knirps* share features such as ectopic activation of JAK/STAT signaling and a differentiation block. We conclude that the embryonic TF *knirps* is an oncogene in eye-antennal imaginal discs

and is crucial for the tumorigenic capacity of the epigenetic tumor under study. Additionally, we found that overexpressing a pro-neural TF leads to reduction of proliferation and suppression of the tumor phenotype.

## Results

$ph^{505}$ mitotic mutant clones were generated using the Flp/FRT system, allowing for specific tumor growth within eye-antennal imaginal discs (from here on referred to as $ph^{505}$-tumor). Mutant clones were fluorescently labeled with GFP, by using the mosaic analysis with a repressible cell marker (MARCM) system (*Wu and Luo, 2006*).

Reduced expression of Ph in the mutant clones compared to tissues carrying FRT19A (neutral) mitotic clones was observed (*Figure 1—figure supplement 1A–B*). Only a small number of larvae carrying $ph^{505}$-tumors reached adulthood (*Figure 1—figure supplement 1C*). The majority of tumors arising in these epithelial tissues display disrupted cell polarity (*Martinez et al., 2009*), which was confirmed using the cell adhesion marker Armadillo (*Figure 1—figure supplement 1D–E*). Additionally, we observed ectopic expression of Matrix metalloprotease 1 (MMP1) (*Figure 1—figure supplement 1F–G*), which is required for matrix degradation and the invasive potential of tumor cells (*Christofi and Apidianakis, 2013*; *Uhlirova and Bohmann, 2006*).

The tumorigenic capacity of PcG mutant tissues has been previously reported (*Classen et al., 2009*; *Martinez et al., 2009*). Here we show for the first time that neoplastic growth of $ph^{505}$ mutant clones in eye-antennal imaginal discs can be rescued by wild-type $ph$ (via UAS-ph) co-expression. The resulting tissues with restored Ph levels harbored smaller clones compared to the tissues containing $ph^{505}$ clones (*Figure 1—figure supplement 1H–I*). To better quantify changes of tumor volume in these tissues we developed an image analysis pipeline (see Materials and methods). This enabled us to concomitantly measure the volume of mutant clones (GFP-labeled) as well as the volume of the tissue. We found that in tissues with $ph^{505}$ clones 46% of the disc was composed of tumor cells, while in the rescue experiment this ratio was reduced to only 7% (*Figure 1—figure supplement 1K–N*). Consistent with these observations, a reduced number of larvae carrying $ph^{505}$-tumors reached adulthood (eclosion rate <20%). In contrast the eclosion rate of $ph^{505}$, UAS-ph pupae (97%) was similar to control flies (99%) (*Figure 1—figure supplement 1J*).

Overall, these results show that the tumorigenic phenotype of $ph^{505}$ tissues is primarily due to the loss of $ph$ since it can be rescued by the expression of the wild-type protein. We had shown previously that tumors with impaired $ph$ do not accumulate genetic instabilities, even when cultivated for prolonged time in adult hosts (*Sievers et al., 2014*). Hence, the observed neoplastic behavior can be solely restricted to the loss of the silencer $ph$ and the ensuing deregulation of the transcriptional state described below.

### $ph^{505}$-tumor cells exhibit global gene expression deregulation

To better understand the molecular consequences of *polyhomeotic* loss-of-function in vivo, we analyzed the transcriptome of $ph^{505}$-tumor cells by RNA sequencing (RNA-seq). We used fluorescence activated cell sorting (FACS) of dissociated eye-antennal imaginal discs to separate GFP-labeled tumor cells from surrounding unlabeled and non-mutant cells (*Martinez et al., 2009*; *Dutta et al., 2013*; *Harzer et al., 2013*). This proved to be an essential step for an accurate diversification of the tumor transcriptome. Transcriptome analysis showed substantial deregulation of gene expression in mutant cells compared to neighboring non-mutant cells. We identified 1337 differentially expressed genes (Benjamini adjusted p-value, padj. <0.01), with 275 genes being upregulated in $ph^{505}$-tumor cells (*Figure 1A*, *Figure 1—figure supplement 2A*, *Figure 1—source data 1*). Furthermore, gene set enrichment analysis revealed that neurogenesis-related genes were mainly downregulated in $ph^{505}$-tumor cells, while genes regulating transcription were upregulated (*Figure 1B*). Since PcG target genes encode crucial developmental regulators, such as TFs (*Simon and Kingston, 2009*), our expression data corroborates its impaired function. Moreover, we observed deregulation of genes involved in tissue development (e.g., GO-terms for genital disc development, imaginal disc development) and enrichment for TFs among the upregulated genes (GO-term sequence-specific DNA binding transcription factor activity) (*Figure 1B*, *Figure 1—source data 2*). The observed global modulation of transcription output is in agreement with PcG proteins constituting a global regulatory system (*Simon and Kingston, 2009*).

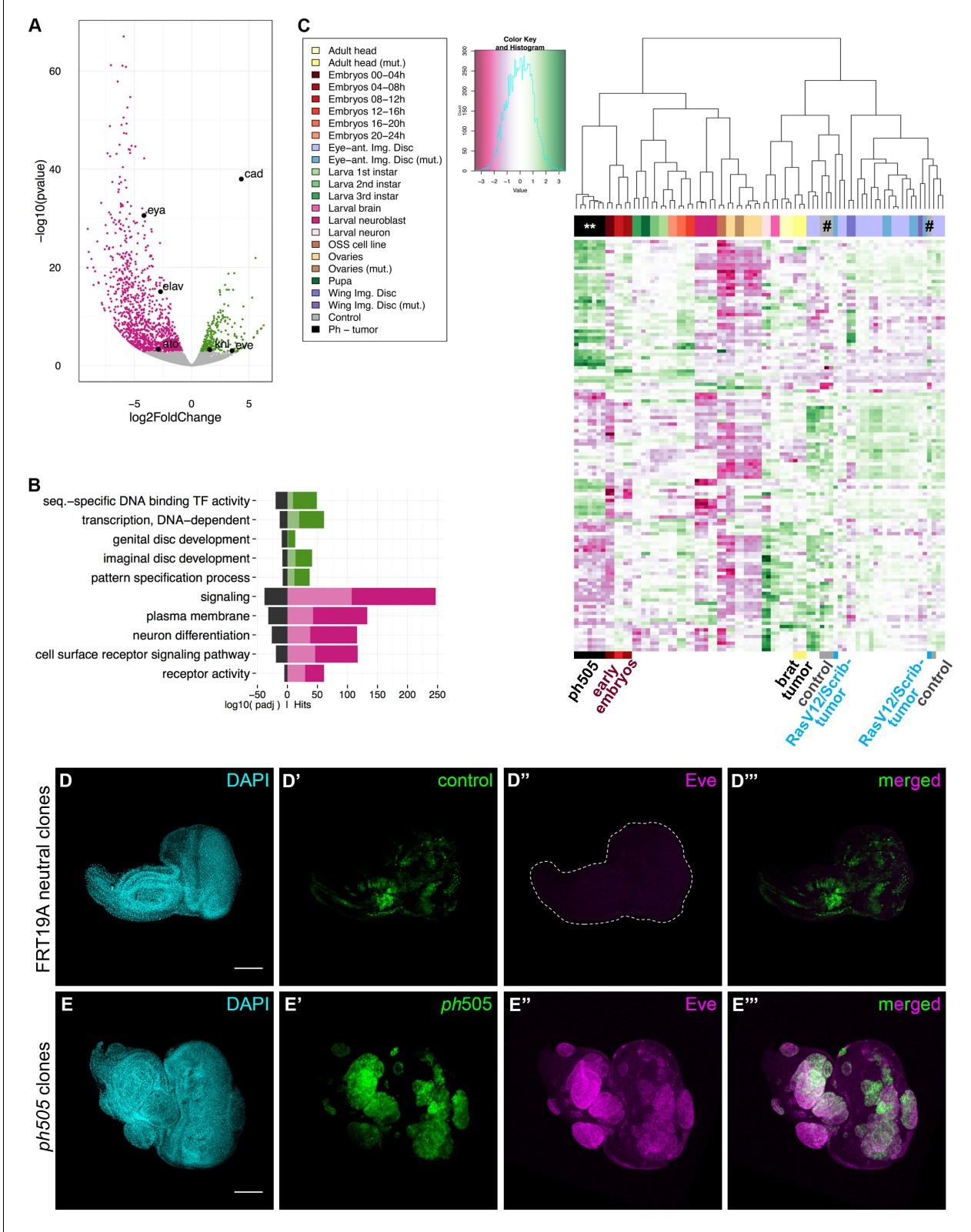

**Figure 1.** Gene expression analysis of *ph*[505]-tumor cells reveals loss of cell identity and acquisition of an embryonic-TF signature. RNA-sequencing was performed in FACS-sorted *ph*[505] tissue samples. Gene expression in tumor cells (GFP[+]) was compared with surrounding wild-type cells (GFP[-]) from the same pool of eye-antennal tissues. Volcano plot in (A) shows 1337 differentially expressed genes: 1062 genes are downregulated (pink) and 275 are upregulated (green) (padj ≤0.01, Benjamini). Names of 6 genes are provided in the volcano plot: *cad, eve, kni, eya, elav, ato*. Representation of

*Figure 1 continued on next page*

*Figure 1 continued*

selected Gene Ontology (GO) terms that were found enriched (for details please check *Figure 1—source data 2*) (B). Green/Pink light bars correspond to the expected number of genes for each category, Green/Pink dark bars correspond to the number of observed hits in our analysis. Black bars correspond to the log10(padj). Pink, downregulated genes; green, upregulated genes. Heatmap and respective dendogram obtained for clustering analysis with 83 samples (*Figure 1—sources data 3* and *4*) (C), showing tumor samples clustering together with early embryos. *ph505*-tumor samples in black (**), control samples (#) in grey and other tumor samples (brat- and RASV12/Scrib- tumors) are highlighted below the clustering. Embryonic transcription factor Even-skipped (Eve) is not detected in FRT neutral clones (D) but is ectopically expressed in *ph505* clones (E). Scale bar corresponds to 100 μm. All microscope images are a maximum intensity projection of all z-stacks acquired for the tissue (DAPI, cyan; GFP MARCM clones, green; antibody staining, magenta). See also *Figure 1—figure supplements 1–4* and *Figure 1—source datas 1–4*.

DOI: https://doi.org/10.7554/eLife.32697.003

The following source data and figure supplements are available for figure 1:

**Source data 1.** List of differentially expressed genes.
DOI: https://doi.org/10.7554/eLife.32697.008

**Source data 2.** Gene Ontology (GO) analysis.
DOI: https://doi.org/10.7554/eLife.32697.009

**Source data 3.** Information relative to samples used for clustering (83 samples).
DOI: https://doi.org/10.7554/eLife.32697.010

**Source data 4.** List of genes used for hierarchical clustering analysis.
DOI: https://doi.org/10.7554/eLife.32697.011

**Source data 5.** List of Genotypes.
DOI: https://doi.org/10.7554/eLife.32697.012

**Figure supplement 1.** Neoplastic growth of *ph* mutant clones in eye-antennal imaginal discs can be rescued by wt *ph* co-expression.
DOI: https://doi.org/10.7554/eLife.32697.004

**Figure supplement 2.** Gene expression analysis of *ph505*-tumor cells reveals loss of cell identity and acquisition of an embryonic-TF signature.
DOI: https://doi.org/10.7554/eLife.32697.005

**Figure supplement 3.** *ph505*-tumor cells do not express differentiation markers.
DOI: https://doi.org/10.7554/eLife.32697.006

**Figure supplement 4.** *ph505*-tumor cells ectopically express embryonic transcription factors.
DOI: https://doi.org/10.7554/eLife.32697.007

## Tumor transcription factor signature reveals loss of larval cell identity and acquisition of embryonic features

TFs are essential to define cell types and are among the main targets of PcG silencing (*Simon and Kingston, 2009*). As such, we decided to focus on the fraction of differentially expressed TF-encoding genes, due also to the enrichment of genes in this category in our RNA-seq dataset. We evaluated which upregulated TFs, and thus the primary response to the *ph* knock-out, could be contributing to the overall deregulation of gene expression observed. Employing the iRegulon tool (*Janky et al., 2014*), predictions based on motif enrichment revealed *caudal, grain* and *knirps* (direct Ph targets in eye discs [*Loubière et al., 2016*]) as TFs putatively responsible for all differentially expressed genes in our RNA-seq dataset. To further reveal the transcriptional identity of *ph505*-tumor cells, we integrated available datasets from the Gene Expression Omnibus (*Edgar et al., 2002*) and compared the gene expression signature of *ph505*-tumors with other tissues and/or developmental stages of *Drosophila*. In total, 83 samples including different tissues and cell types (namely ovaries, larval brain, adult head, wing disc, eye-antennal disc, larval neurons and larval neuroblasts) and developmental stages (embryo, larva and pupa) were considered for the analysis (*Figure 1—source data 3*). Specifically, we compared 124 differentially expressed genes involved in transcriptional regulation (GO0006355) (*Figure 1—source data 4*). Strikingly, hierarchical clustering of 83 transcriptome samples showed that *ph505*-tumor cells clustered close to samples from early embryonic developmental stages (*Figure 1C* and *Figure 1—figure supplement 2B*). As expected, our RNA-seq control samples (neighboring unlabeled cells) clustered with wild-type eye-antennal imaginal disc transcriptomes. This result might reflect the re-establishment of an earlier developmental program in *ph505*-tumors as a consequence of a reprogrammed epigenomic state. Additionally, this is not a general feature shared by all tumors, as depicted by other tumor types (i.e. *brat* [*Jüschke et al., 2013*] and *RasV12/scrib-* tumors [*Atkins et al., 2016*]) not clustering with embryos.

We hypothesize that the clustering of $ph^{505}$-tumor cells with early embryos was not only the result of the ectopic expression of embryonic TFs in the $ph^{505}$-tumor cells, but also due to reduced expression of the TFs, which characterize differentiated tissues. Downregulation of neurogenesis-related genes suggests that these tumor cells may be unable to differentiate, losing cell fate markers and their normally established identity. We confirmed the downregulation of neurogenesis-related markers at the protein level for ELAV (Embryonic Lethal Abnormal Vision), which is normally expressed in the differentiated neuronal cells that make up the eye imaginal disc (*Figure 1—figure supplement 3A–B*) and for Eya (Eyes absent, *Figure 1—figure supplement 3C–D*). This supports previous findings that suggested that neoplastic *Drosophila* epithelial cells reverse their developmental commitments and switch to primitive cell states (*Khan et al., 2013*). In this particular report, the switch in the eye primordium was shown to be Homothorax (Hth)-dependent (*Khan et al., 2013*). Conversely, in our RNA-seq dataset *hth* was downregulated and at the protein level we confirmed that Hth is not ectopically expressed in the $ph^{505}$ clones (*Figure 1—figure supplement 2E–F*). Thus, our study reveals that $ph^{505}$-tumors do not depend upon ectopic expression of Hth to keep cells in a non-differentiated state and support neoplastic growth.

The similarity of the $ph^{505}$-tumor TF signature with *Drosophila* early embryos was reinforced by confirmation of the presence of embryonic-TF misexpression across tumor-tissue samples. We performed immunostaining for additional embryonic TFs, namely Even-skipped, Abdominal-B and Caudal, and observed ectopic expression of these proteins specifically in mutant clones (*Figure 1D–E* and *Figure 1—figure supplement 3A–D*). Overall, these results suggest that $ph^{505}$-tumor cells previously committed to a neurogenesis-related path switch their cell fate as they fail to differentiate during the process of tumorigenesis due to the modulation of the transcriptional regulatory program of the cells.

## TFs as key regulators of tumorigenesis – candidate-hit validation in vivo

In order to pinpoint key regulators of tumorigenesis in $ph^{505}$-tumors, we performed an *in vivo* screen for a subset of selected candidates. Among all the TFs upregulated in $ph^{505}$-tumor cells, we chose, based on literature search, 24 to assess their importance in promoting tumorigenic potential in these cells. Our approach to test the ability of candidate genes playing a key role in $ph^{505}$-tumorigenesis was to combine generation of $ph^{505}$-tumor clones with RNAi-mediated knock-down (KD) of a target of interest within these clones. We compared effects of RNAi to the baseline $ph^{505}$ neoplastic phenotype and observed that some RNAi lines targeting TFs (in $ph^{505}$ clones) resulted in a strong increase in viability (close to 90–100%, for example *cad, drm, kni, bgcn*), while others did not change or only slightly changed pupal viability (*Figure 2A*). We further characterized 6 RNAi lines: *crocodile* (*croc*), *lateral muscles scarcer* (*lms*), *caudal* (*cad*), *drumstick* (*drm*), *knirps* (*kni*) and *benign gonial cell neoplasm* (*bgcn*) (*Figure 2B–H*), which showed significant differences in eclosion rate in comparison to flies carrying $ph^{505}$ clones (*Figure 2B* and *Figure 2—figure supplement 1A*). The eclosion rate for three of these perturbations (drm-, kni- and bgcn-KD) reached similar levels as control flies (carrying FRT19A neutral clones) and rescue experiment ($ph^{505}$, UAS-ph). By quantifying tumor volumes relative to tissue size of the six above-mentioned perturbations, we observed that only cad-, drm-, kni- and bgcn-KD showed a significant difference compared to the baseline of 46% tumor volume in the $ph^{505}$ condition (14, 13, 5 and 14% tumor volume, respectively). In addition, the two perturbations (croc- and lms-KD) that did not show a significant effect on tumor volume were also those with less remarkable differences in eclosion rate (*Figure 2B* and *Figure 2—figure supplement 1B–D*). These results suggest that higher eclosion rate is a good approximation for decreased tumor volume. Furthermore, the tissue volume of the eye-antennal imaginal disc in the drm-, kni- and bgcn-KD conditions was closer to the control tissue volume than the $ph^{505}$ condition (*Figure 2—figure supplement 1C*).

From all the RNAi conditions tested, kni-KD in $ph^{505}$ mutant clones showed the most striking decrease in tumor volume (9.2 fold decrease), similar to the rescue experiment ($ph^{505}$, UAS-ph) (*Figure 2B*). Additionally, the phenotype of adult eyes of this genotype suggests a recovery of the differentiation program (*Figure 2—figure supplement 2A–D*). This is supported by immunostaining against ELAV showing that it is no longer disrupted when expression of *kni* is blocked in $ph^{505}$ (*Figure 2—figure supplement 2E*). Altogether, this shows that the differentiation block observed in $ph^{505}$-tumors is prevented upon reducing the level of *knirps* expression by RNAi KD. We confirmed that a second, independent RNAi line against *kni* also led to a significant decrease of tumor volume

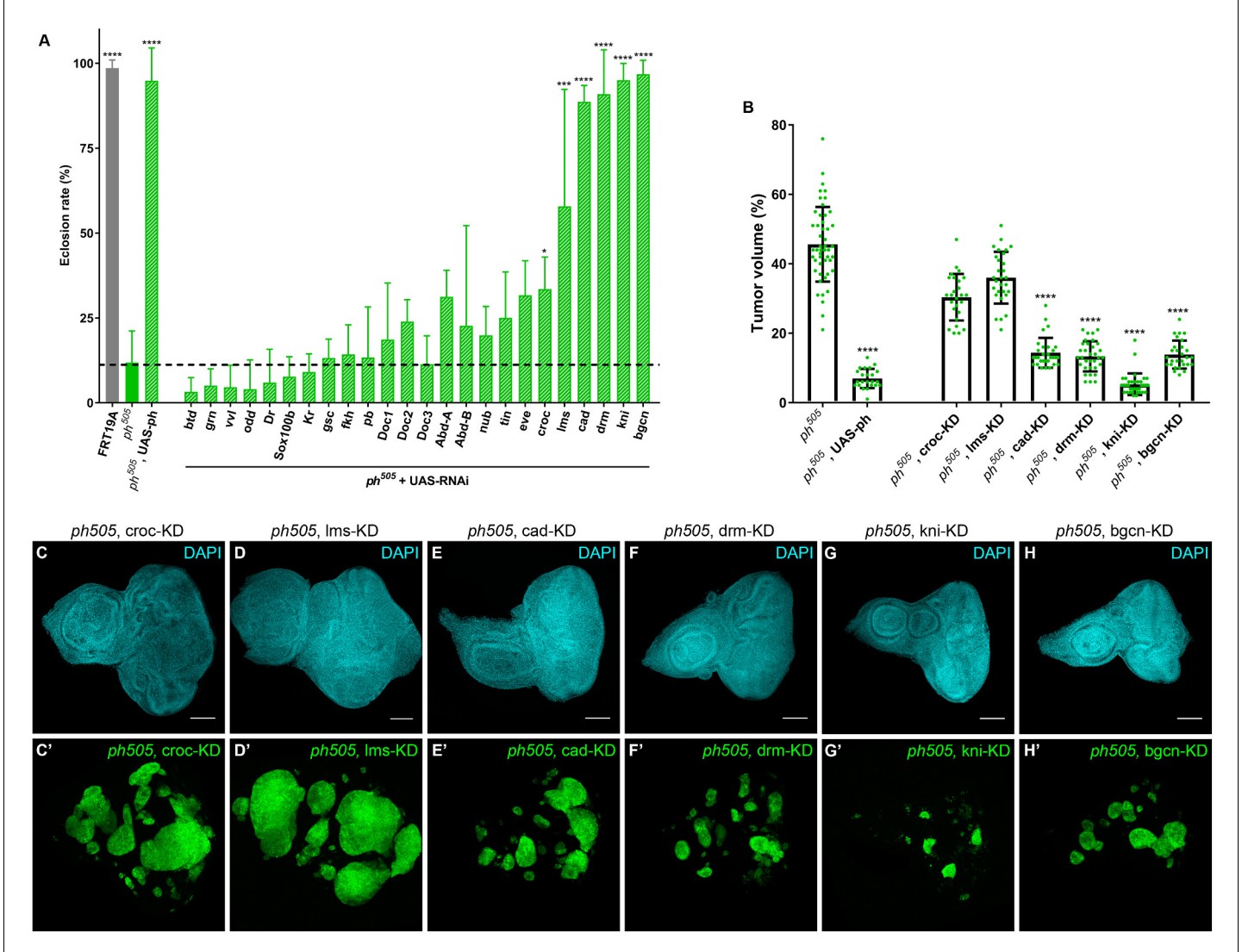

**Figure 2.** KD of embryonic TFs in $ph^{505}$-tumors can increase the viability of the flies and reduce tumor volume. Eclosion rates (%) for FRT19A control, $ph^{505}$ and for specific KD in $ph^{505}$ background ($ph^{505}$ +UAS RNAi). Fly stocks carrying RNAi constructs were used for the KD of 24 TFs upregulated in $ph^{505}$-tumor cells (**A**). Dashed line represents the mean eclosion rate for $ph^{505}$ larvae. Number of larvae analyzed: FRT19A N = 163; $ph^{505}$ N = 784; $ph^{505}$, UAS-ph N = 48; for each TF-KD in $ph^{505}$ background: btd- N = 329, grn- N = 256, vvl- N = 250, odd- N = 108, Dr- N = 96, Sox100b- N = 304, Kr-N = 309, gsc- N = 177, fkh- N = 231,pb- N = 110, Doc1- N = 337, Doc2- N = 128, Doc3- N = 297, AbdA- N = 119, AbdB- N = 129, nub- N = 216, tin-N = 54, eve- N-117, croc- N = 355, lms- N = 58, cad- N = 137, drm- N116, kni- N = 340, bgcn- N = 158. Targets whose KD induced a significantly different eclosion rate compared with $ph^{505}$ were further characterized at the tumor volume level (**B**). Number of tissues analyzed *per* condition: $ph^{505}$ N = 50; $ph^{505}$, UAS-ph N = 23; $ph^{505}$, croc-KD N = 26; $ph^{505}$, lms-KD N = 29; $ph^{505}$, cad-KD N = 30; $ph^{505}$, drm-KD N = 33; $ph^{505}$, kni-KD N = 35; $ph^{505}$, bgcn-KD N = 28. Percentage of tumor volume is significantly reduced in 4 out of 6 perturbations compared with $ph^{505}$ alone (46%): 14% in $ph^{505}$, cad-KD; 13% in $ph^{505}$, drm-KD; 5% in $ph^{505}$, kni-KD; and 14% in $ph^{505}$, bgcn-KD. $ph^{505}$, UAS-ph condition leads to reduction of tumor volume to 7% and is shown here as a control of the rescue phenotype. Examples of eye-antennal imaginal discs from six different genotypes used for quantification of tumor volume (**C–H**). Scale bar corresponds to 100 μm. All microscope images are a maximum intensity projection of all z-stacks acquired for the tissue (DAPI, cyan; GFP MARCM clones, green). Data (**A–B**) are represented as mean ± SD. Statistics: ****p<0.0001; ***p<0.001; *p<0.05. See also ***Figure 2—figure supplement 1*** and ***2***.

DOI: https://doi.org/10.7554/eLife.32697.013

The following figure supplements are available for figure 2:

**Figure supplement 1.** KD of embryonic TFs in $ph^{505}$-tumors can increase the viability of the flies and reduce tumor volume.
DOI: https://doi.org/10.7554/eLife.32697.014
**Figure supplement 2.** KD of embryonic TFs in $ph^{505}$-tumors can increase the viability of the flies and reduce tumor volume.
DOI: https://doi.org/10.7554/eLife.32697.015

(14% of tumor volume vs. 46% in $ph^{505}$) and an increase in eclosion rate (85%) (*Figure 2—figure supplement 2F–J*). We can thus minimize the chance that the effects observed using either kni-RNAi were due to off-target effects.

## knirps-KD reduces $ph^{505}$ tumorigenic capacity

We characterized the tumorigenic potential of $ph^{505}$-tumors and $ph^{505}$, kni-KD by conducting transplantation assays (*Rossi and Gonzalez, 2015*) of these tissues into the abdomen of adult host flies (*Figure 3A–D*). In the case of the $ph^{505}$-tumors the percentage of tumor-bearing hosts increased from 40% to 60%, from the first week to subsequent weeks after transplantation indicating hyperproliferation of the transplanted tissues (*Figure 3A*). The tumorigenic potential of $ph^{505}$-transplanted tissue was already detected on day seven after transplantation (*Figure 3B*). By contrast, when transplanting $ph^{505}$, kni-KD clones we did not observe any tumors in the host flies within the first three weeks. Even after up to 5 weeks post-transplantation, we could only find a single fly with GFP⁺ tissue overgrowth (*Figure 3C–D*). Our data demonstrate that the TF Knirps plays a crucial role in tumorigenesis of $ph^{505}$-tumors given that kni-KD in these tissues not only led to a reduction of tumor volume but also the remaining clones were not able to proliferate in the host fly abdomen.

Evasion of apoptosis is one of the hallmarks of cancer (*Hanahan and Weinberg, 2000*). As we observed a significant reduction of tumor volume upon depleting *kni* in $ph^{505}$ mutant cells, we hypothesized that kni-KD could trigger cell death of tumor cells. We blocked apoptosis within mutant clones (via expression of anti-apoptotic protein p35 [*Hay et al., 1994*]). Levels of apoptosis as assessed by immunostaining against Death caspase-1 (Dcp-1) confirmed a decrease in apoptosis in tissues where p35 was expressed in $ph^{505}$, kni-KD clones (*Figure 3E–F* and *Figure 3—figure supplement 1A–B*). However, we observed that blocking apoptosis in $ph^{505}$, kni-KD clones was not sufficient to revert the anti-oncogenic effects of kni-KD (*Figure 3G–H* and *Figure 3—figure supplement 1C–F*). Furthermore, the tumor volume of $ph^{505}$, kni-KD, UAS-p35 was similar to $ph^{505}$, kni-KD (*Figure 3G* and *Figure 3—figure supplement 1D–F*). We also tested the effect of this particular RNAi line in the context of neutral clones generated with the same driver as for $ph^{505}$-tumors. These FRT19A, kni-KD flies showed neither difference in eclosion rate nor in the adult eye phenotype, compared to control flies (*Figure 3E–F* and *Figure 3—figure supplement 1C*). This suggests that the RNAi line targeting *kni* does not per se affect eye-antennal imaginal disc development.

## Ectopic expression of *knirps* is sufficient to drive tumorigenesis

Ectopic expression of cell fate-specifying TFs was recently shown to lead to the formation of epithelial cysts (*Bielmeier et al., 2016*). Cyst formation in wing and eye imaginal discs represents a response to cell fate mis-specification, compromising tissue integrity and potentially promoting precancerous lesions. We thus assessed the effect of ectopic *kni* expression in eye-antennal imaginal discs by generating mitotic clones using again the eyFlp system. We observed that FRT19A clones expressing *kni* (*Figure 4A*) displayed a more pronounced round shape in comparison to the notchyshape of FRT19A neutral clones (*Figure 1—figure supplement 1A*). Additionally, ectopic *kni* expression compromised the viability of the flies, evidenced by an eclosion rate of only 35% (*Figure 4B*), and the defective development of the adult eye structures (*Figure 4C*). Furthermore, we confirmed that ectopic expression of *kni* leads to the formation of cysts (*Figure 4D*) and thus interferes with epithelial polarity (*Figure 4D* and *Figure 5—figure supplement 1A*).

To test if ectopic expression of *kni* alone is sufficient to drive tumorigenesis we conducted transplantations of eye-antennal imaginal disc tissues ectopically expressing *kni*. We observed that *knirps* is sufficient to generate tumors in the host flies, visible 3 weeks after transplantation (ranging from 15–50%, from week 3 to 5 after transplantation respectively) (*Figure 4E–F*). Our data suggest that ectopic expression of *knirps* interferes with the normal course of development and that *knirps* is a new oncogene, possibly acting in a context/tissue-dependent manner.

## *Knirps* activates JAK/STAT pathway and blocks cellular differentiation

Since *knirps*-KD alone was sufficient to reduce the tumorigenic potential of $ph^{505}$-tumors, we hypothesized that some features of clones ectopically expressing *knirps* in a wild-type context could resemble $ph^{505}$-tumor clones. We therefore evaluated the activation of signaling pathways in this context. We observed that of the JNK, JAK/STAT and Notch signaling pathways (all are activated in $ph^{505}$-

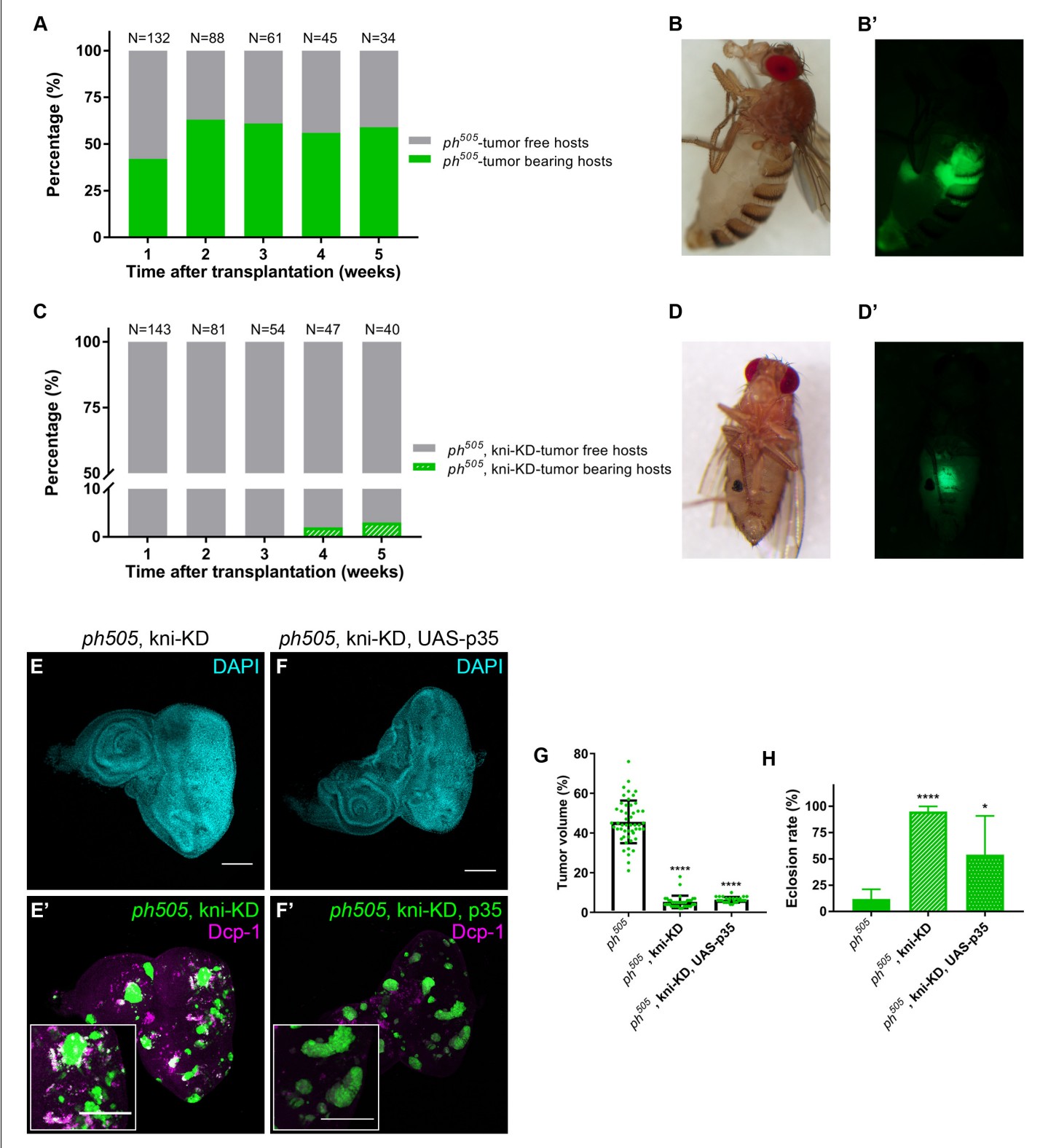

**Figure 3.** kni-KD in $ph^{505}$-tumors has anti-oncogenic effects Percentage of $ph^{505}$ tumor-bearing hosts over time ranges from 40–60% (A). Female host shows abdominal tumor growth one week after transplantation (B–B'). Percentage of $ph^{505}$, kni-KD tumor-bearing hosts is shown over time and ranges from 0% to 3% (C). One single host with a tumor was observed on week 4 (C). Brightfield (D) and fluorescence images (D') of the tumor-bearing host on day 35. 'N' represents the total number of hosts analyzed *per* time point. Green bars represent the percentage of hosts with visible tumors, while grey

*Figure 3 continued on next page*

*Figure 3 continued*

bars represent the percentage of hosts without tumors. Levels of apoptosis (Dcp-1) are shown for $ph^{505}$, kni-KD (E–E') and $ph^{505}$, kni-KD, UAS-p35 (F–F'). Higher magnification insets are shown in E' and F'. Percentage of tumor volume (G). Number of tissues analyzed *per* condition: $ph^{505}$ N = 50; $ph^{505}$, kni-KD N = 35; $ph^{505}$, kni-KD, UAS-p35 N = 27. Percentage of tumor volume is significantly reduced in the two conditions compared with $ph^{505}$ alone (46%): 5% in $ph^{505}$, kni-KD; and 6% in $ph^{505}$, kni-KD, UAS-p35. Eclosion rate for same conditions (H). Number of larvae analyzed: $ph^{505}$ N = 784; $ph^{505}$, kni-KD N = 340; $ph^{505}$, kni-KD, UAS-p35 N = 48. Scale bar corresponds to 100 μm. All microscope images are a maximum intensity projection of all z-stacks acquired for the tissue (DAPI, cyan; GFP MARCM clones, green). Data (G–H) are represented as mean ± SD. Statistics: ****p<0.0001; *p<0.05. See also *Figure 3—figure supplements 1*

DOI: https://doi.org/10.7554/eLife.32697.016

The following figure supplement is available for figure 3:

**Figure supplement 1.** kni-KD in $ph^{505}$-tumors has anti-oncogenic effects.

DOI: https://doi.org/10.7554/eLife.32697.017

tumors [*Classen et al., 2009*; *Martinez et al., 2009*; *Beira et al., 2018*]), only JAK/STAT was ectopically activated, particularly in *knirps* cyst-like clones (*Figure 5A–C*). This observation suggests that ectopic expression of *knirps* alone is sufficient to activate the JAK/STAT pathway in mitotic clones. Hence, ectopic activation of the other signaling pathways in $ph^{505}$-tumors is likely attributable to other factors regulated by Ph and independent of *knirps*.

In light of the compromised eye development seen in kni-ectopic flies, and suggestions that the JAK/STAT pathway needs to be switched off to allow differentiation (*Amoyel and Bach, 2012*), we investigated the expression of a number of neurogenesis-related markers in kni-ectopic eye-antennal imaginal discs. Similarly to what we observed with $ph^{505}$ clones, ELAV expression was disrupted in kni-expressing cyst-like structures, as shown in *Figure 5D*, as well as Eya (*Figure 5E*), without ectopic activation of Hth (*Figure 5F*). These observations are thus in agreement with the hypothesis that *knirps* alone is sufficient to initiate tumorigenesis.

Our data argue in favor of a role for JAK/STAT in contributing to the differentiation block in $ph^{505}$ and kni-ectopic tumors. We decided to block this pathway in $ph^{505}$-tumors ($ph^{505}$, dome$^{\Delta CYT}$) and examine cellular differentiation in these eye-antennal imaginal discs. Upon blocking JAK/STAT in $ph^{505}$-tumors, we observed that ELAV expression is re-established almost to a normal situation, even in the presence of clones (*Figure 5—figure supplement 1B* in comparison to *Figure 1—figure supplement 3B*). Moreover, the viability of these flies is increased, close to normal levels (*Figure 5—figure supplement 1C*, eclosion rate 85%) and some adult flies presented eye structures similar to *wt* individuals *Figure 5—figure supplement 1D*).

## Overexpression of a pro-neural TF in $ph^{505}$-clones suppresses the tumorigenic phenotype

Blockage of normal differentiation appears to be a common feature between $ph^{505}$ and kni-ectopic tumors in eye tissues, suggesting that *kni* expression in the $ph^{505}$-tumors contributes to the differentiation defects observed (*Figure 1—figure supplement 3* and *Figure 2—figure supplement 2C*). Hence, we expected that apart from the knock-down of an embryonic TF with tumorigenic capacity, forcing differentiation of tumor cells could restrain the tumorigenic phenotype.

*Atonal (ato)*, encoding a pro-neural TF, was previously shown to have an anti-oncogenic role in the fly retina, where it instructs tissue differentiation (*Bossuyt et al., 2009*). Notably, *ato* is also among our downregulated set of genes (padj. <0.01). We ectopically expressed *ato* in $ph^{505}$ clones, which led to the rescue of the phenotype by a reduction of the tumor volume from 46% baseline to 3% and an increase in the eclosion rate from 12% to 84% (*Figure 6A–C* and *Figure 6—figure supplement 1A–E*). Hence, expression of *ato* in $ph^{505}$ clones was sufficient to restore the normal pattern of differentiation of this tissue, as confirmed by the expression of ELAV (*Figure 6D* and *Figure 6—figure supplement 1G*). Indeed, also the eye phenotype of the hatched flies resembled the phenotype of wild-type flies (*Figure 6—figure supplement 1F*). We then asked whether these effects can be attributed to the capacity of *atonal* in preventing proliferation, as previously shown in a different tumor model (*Bossuyt et al., 2009*). To test this hypothesis, we assessed levels of phospho-histone H3 (pH3) as a measure of proliferation (*Figure 6E–G* and *Figure 6—figure supplement 2*). Quantitative analysis showed an overall increase in proliferation levels in $ph^{505}$-tumor tissues in comparison to control tissues. This was largely due to an increase of proliferative cells outside of $ph^{505}$ clones

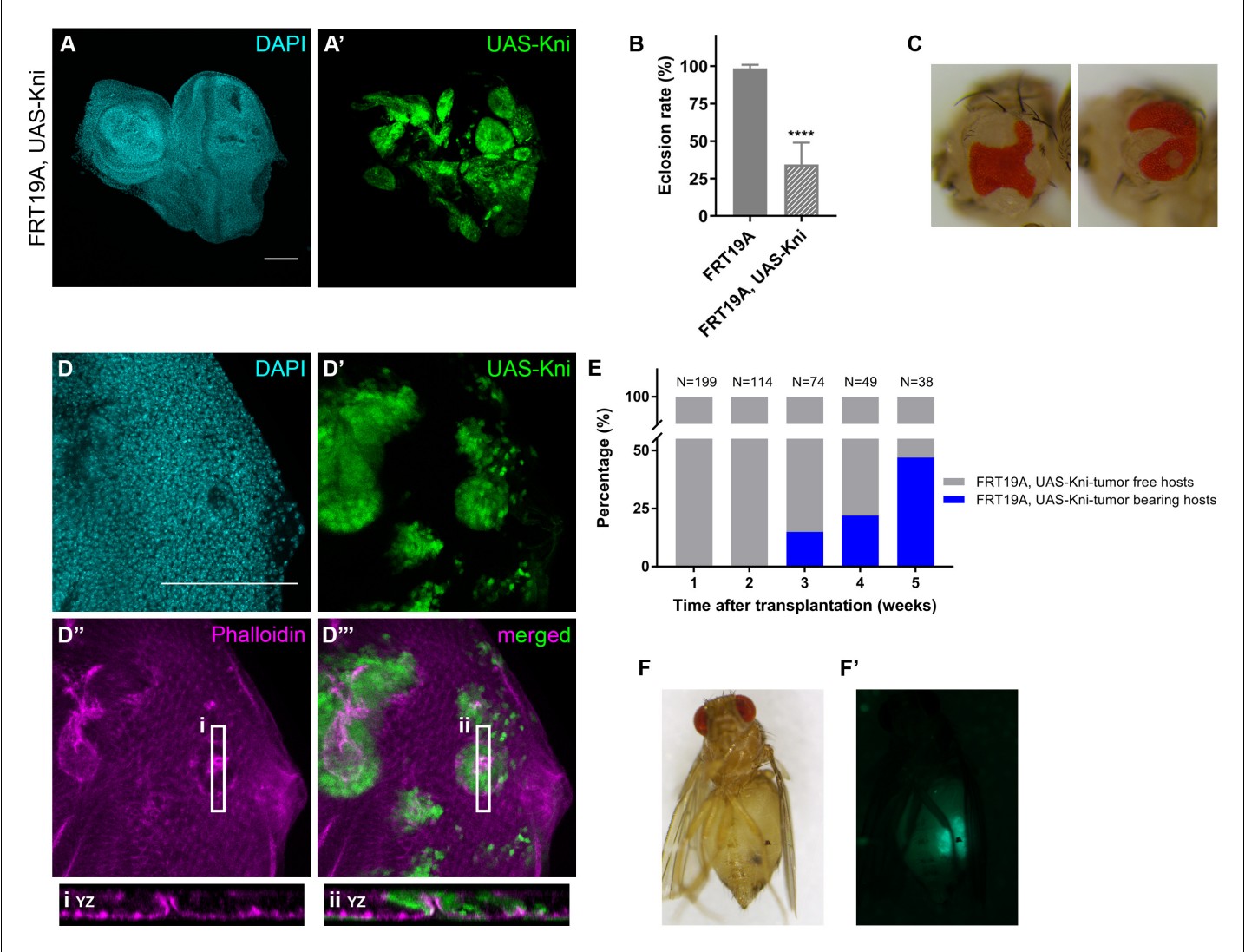

**Figure 4.** Knirps misexpression induces cyst formation in imaginal discs and is sufficient to drive tumorigenesis. Ectopic expression of *knirps* in eye-antennal imaginal discs shows a distinct pattern of clones (A–A'). Eclosion rate (B) for FRT19A and FRT19A, UAS-Kni. Ectopic expression of *kni* in these particular tissues compromises the viability of the flies. Number of larvae analyzed: FRT19A N = 163; FRT19A, UAS-Kni N = 441. Ectopic expression of *knirps* in eye-antennal discs compromises normal eye development as assessed by adult eye structures (C). Formation of cysts in tissues carrying ectopic expression of *kni* (D–D'), highlighted by middle sections (five middle Z-stacks) with Phalloidin staining (D''). Insets (i and ii) for orthogonal views (YZ) are shown and cutting plane is depicted in (D''–D'''). Assays of transplantation were performed with tissues carrying ectopic expression of *kni* (E). Tumor-bearing hosts were observed 3 weeks after transplantation (<25% of hosts). In week 5, almost 50% of the hosts that survived carried GFP+-tumor tissue. Blue bars represent the percentage of hosts with tumors, while grey bars show percentage of hosts without tumor. Tumor-bearing host on week five is shown in (F) and tumor fluorescence in (F'). Scale bar corresponds to 100 µm. All microscope images are a maximum intensity projection of all z-stacks acquired for the tissue, except where otherwise stated (DAPI, cyan; GFP MARCM clones, green; antibody staining, magenta). Data (B) are represented as mean ± SD. Statistics: ****p<0.0001. See also *Figure 5—figure supplement 1*.
DOI: https://doi.org/10.7554/eLife.32697.018

(*Figure 6—figure supplement 2C*). The analysis also showed a decrease in pH3+ cells inside clones co-expressing *ph505* and *atonal* (in comparison to *ph505* clones) (*Figure 6G*). Thus, *atonal* antagonizes *ph* tumor growth by counterbalancing proliferation, ultimately leading to a reduction of tumor burden and to a normal eye differentiation pattern.

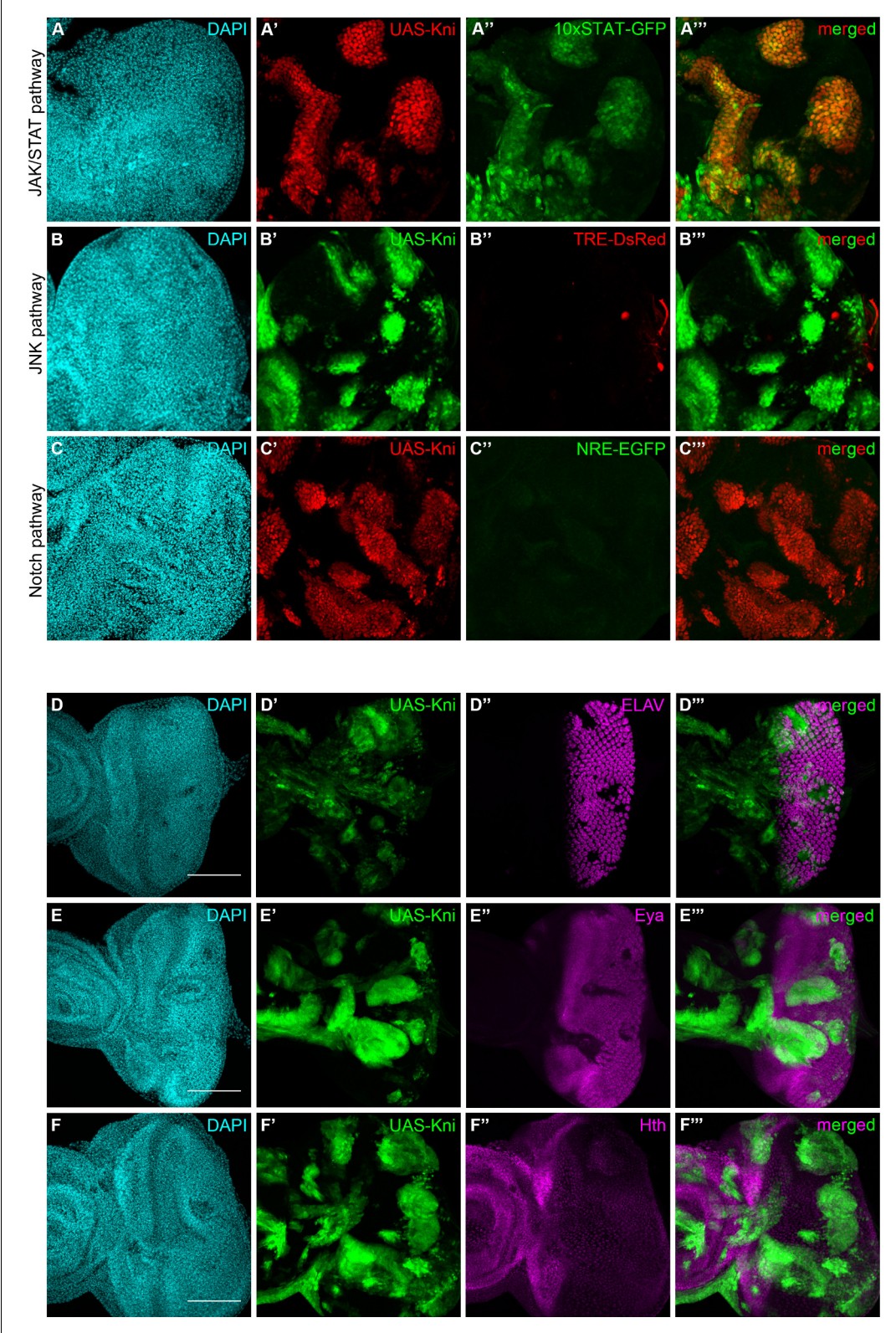

**Figure 5.** Knirps misexpression induces JAK/STAT activation and compromises differentiation. Activity of JAK/STAT, JNK and Notch signaling pathways in the context of kni-ectopic expression in eye-antennal discs was assessed by evaluating the expression of the respective activity reporters: 10x STAT-GFP (**A**, RFP MARCM clones in red, JAK/STAT reporter in green), TRE-DsRed (**B**, GFP MARCM clones in green, JNK reporter in red) and NRE-EGFP (**C**, RFP MARCM clones in red, Notch reporter in green). Ectopic expression of STAT is specifically observed in kni cyst-like clones (**A'–A'''**). JNK and Notch

*Figure 5 continued on next page*

*Figure 5 continued*

pathways are not ectopically activated in UAS-Kni clones. Differentiation of eye progenitors is compromised as observed by ELAV and Eya protein expression (D–E). Normal ELAV and Eya protein expression is interrupted in the presence of Kni ectopic clones, particularly in cysts (D''' and E'''), without ectopic expression of Hth (F). Scale bar corresponds to 100 μm. All microscope images are a maximum intensity projection of all z-stacks acquired for the tissue (DAPI, cyan; GFP MARCM clones, green, unless otherwise stated; antibody staining, magenta). See also *Figure 5—figure supplement 1*.

DOI: https://doi.org/10.7554/eLife.32697.019

The following figure supplement is available for figure 5:

**Figure supplement 1.** Knirps misexpression induces JAK/STAT activation and compromises differentiation.
DOI: https://doi.org/10.7554/eLife.32697.020

## Discussion

### Loss of an epigenetic regulator leads to acquisition of an embryonic and oncogenic gene signature

With the analysis of a *ph* mutant transcriptome we highlight the complexity of disrupting global gene expression programs and, with that, newly established transcriptional dependencies. Previous approaches of generating PcG-negative transcriptomes investigated gene expression of mutant cells that were deprived from contact with non-mutant cells (using the cell lethal system), which was then compared with wild-type discs composed of neutral clones (*Loubière et al., 2016*; *Bunker et al., 2015*). In contrast, we set out to compare $ph^{505}$ mutant cells with their surrounding wild-type cells to gain potential additional information by taking into account non-autonomous growth effects previously reported (*Feng et al., 2011*). The RNA-seq dataset presented here reveals enrichment for TFs in the upregulated gene set. It also indicates that tumor cells fail to differentiate, supported by the downregulation of neural-cell fate markers characteristic of this tissue and by the upregulation of embryonic TFs. This is also highlighted by the clustering of the TF-signature of $ph^{505}$-tumors with embryonic stages of *Drosophila* development. Moreover, we also found several Hox genes in our set of upregulated genes (e.g., *Antp, Ubx, Abd-A, Abd-B*), which are classical embryonic PcG-targets shown to be important in oncogenesis.

Although not regarded as a traditional hallmark of cancer (*Hanahan and Weinberg, 2000*), a key event in tumorigenesis is the perturbation of normal cell fate (*Gonda and Ramsay, 2015*). Re-expression of particular embryonic genes in an aberrant spatial-temporal pattern could contribute to oncogenesis by maintenance of a more embryonic state through the activation of anti-apoptotic pathways or suppression of differentiation (*Shah and Sukumar, 2010*). For example, re-establishment of an earlier developmental program has been proposed in human pediatric gliomas that frequently have mutations in histone H3 lysine 27 (H3K27M) and compromised PRC2 function (*Funato et al., 2014*; *Wainwright and Scaffidi, 2017*).

Since several classic TFs with important functions during embryogenesis are among the upregulated genes in the $ph^{505}$-tumor transcriptome, we subsequently blocked their expression and showed for some TFs their potential to rescue the *ph* knock-out phenotype and reduce tumor growth. Quantitative measurements of tumor volume in various conditions ensured the reproducibility of the data, excluding an observer bias. The observed effects of TF-KD on eclosion rate and tumor volume did not necessarily correlate with the genes that are direct targets of Ph silencing in eye discs. This is illustrated for example by the strong effects of bgcn-KD that has not been identified as a direct Ph target (*Loubière et al., 2016*).

These observations on TFs are particularly important since transcription has a direct influence on the balance between proliferation and differentiation. Furthermore, when transcriptional regulators (TFs, co-regulators or epigenetic modifiers) are misregulated, differentiation is blocked and pre-cancer cells can proliferate (*Gonda and Ramsay, 2015*).

### An embryonic nuclear hormone receptor as new oncogene

*kni* is a gap gene involved in the subdivision of the embryo anterior-posterior axis that can function as an activator (*Langeland et al., 1994*) or a repressor (*Pankratz et al., 1990*). Besides its classic function in embryonic development, *kni* is subsequently also required for vein formation in wing imaginal discs (*Lunde et al., 2003*). We show that KD of *knirps* in $ph^{505}$-tumors is sufficient to reduce

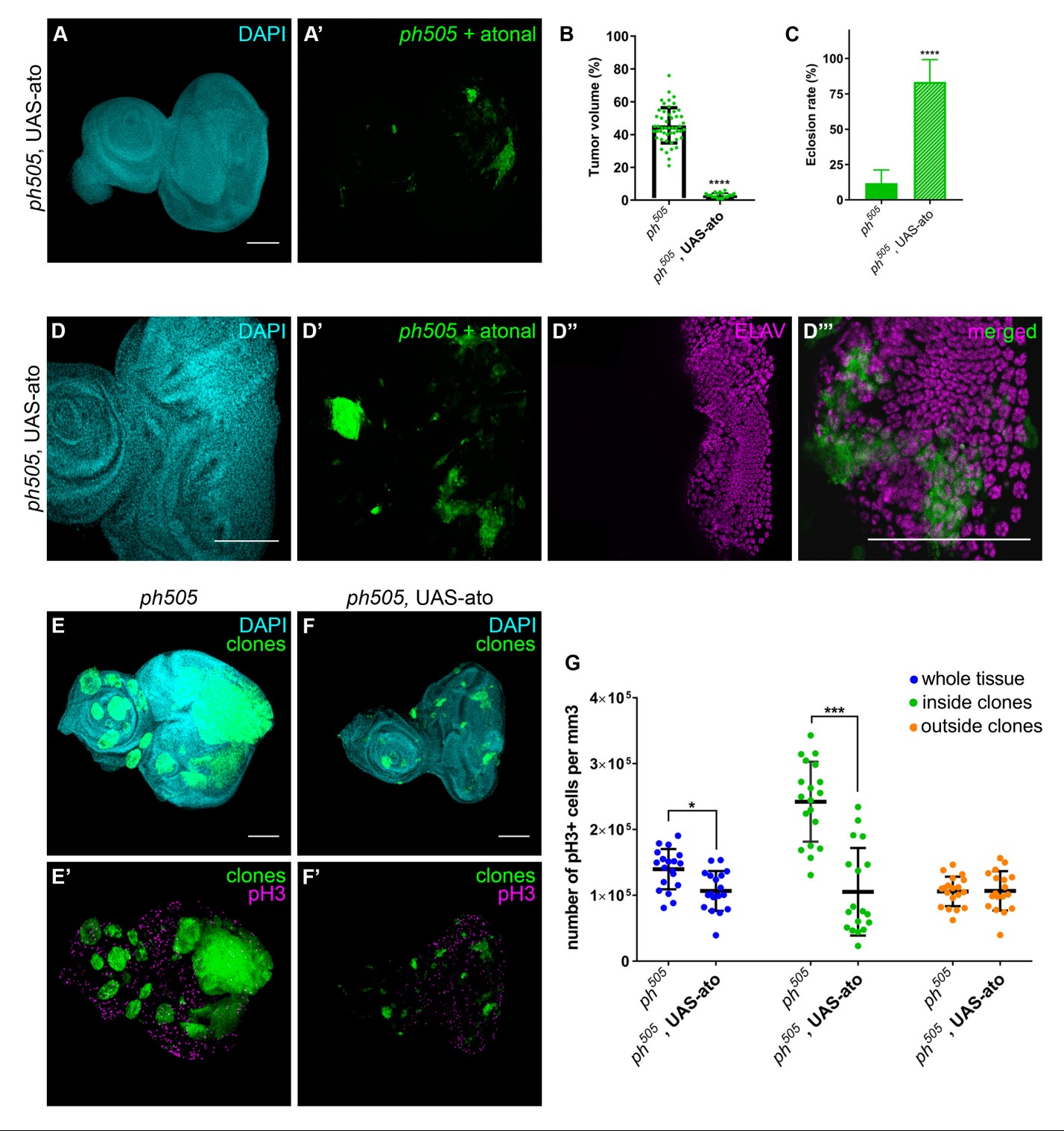

**Figure 6.** Overexpression of pro-neural TF in $ph^{505}$-clones suppresses tumor phenotype. Forcing differentiation of $ph^{505}$ cells by ectopically expressing pro-neural TF *atonal* leads to a reduction of tumor volume (A–B). Number of tissues analyzed: $ph^{505}$ N = 50; $ph^{505}$, UAS-ato N = 23. On average, tumor volume is reduced to 3% of the eye-antennal imaginal disc volume (B). Eclosion rate of larvae of $ph^{505}$, UAS-ato genotype is increased and comparable to FRT19A neutral clones' genotype (C). Number of larvae analyzed: $ph^{505}$ N = 784; $ph^{505}$, UAS-ato N = 95. ELAV protein expression in $ph^{505}$ clones expressing UAS-ato (D). Proliferation levels (phospho-histone H3, pH3) upon overexpressing *ato* in $ph^{505}$-cells (E–F). Quantitative analysis of pH3+ cell numbers in $ph505$ and $ph505$, UAS-ato (G). These values were normalized to the respective volumes, such as 'whole tissue' data was normalized to the total tissue volume, 'inside clones' normalized to volume taken by GFP+ clones and 'outside clones' normalized to the volume of tissue that is GFP-.
*Figure 6 continued on next page*

*Figure 6 continued*

Number of tissues analyzed: $ph^{505}$ N = 19; $ph^{505}$, UAS-ato N = 18. Scale bar corresponds to 100 μm. All microscope images are a maximum intensity projection of all z-stacks acquired for the tissue (DAPI, cyan; GFP MARCM clones, green; antibody staining, magenta). Data (**B–C, G**) are represented as mean ± SD. Statistics: ****p<0.0001, ***p<0.001, *p<0.05. See also *Figure 6—figure supplements 1–2*.
DOI: https://doi.org/10.7554/eLife.32697.021
The following figure supplements are available for figure 6:

**Figure supplement 1.** Overexpression of pro-neural TF in $ph^{505}$-clones suppresses tumor phenotype.
DOI: https://doi.org/10.7554/eLife.32697.022
**Figure supplement 2.** Overexpression of pro-neural TF in $ph^{505}$-clones prevents cells from proliferating.
DOI: https://doi.org/10.7554/eLife.32697.023

---

tumor volume by 90%. It also reduces the tumorigenic capacity of $ph^{505}$-tumors, as assessed by a transplantation assay. Misexpression of TFs in imaginal discs and formation of cysts has been suggested to be an indicator of precancerous lesions (*Bielmeier et al., 2016*). Here we show that ectopic expression of *knirps* in eye-antennal imaginal discs leads to the formation of cysts and is sufficient to recapitulate the phenotypic tumor appearance. Moreover, we believe that this TF, with its dual regulatory role, could activate or repress other genes and thus form a regulatory circuit that is beneficial for tumor initiation and progression.

Inducing a cell fate switch can be achieved by forcing expression of a TF that can activate the transcriptional network of the resulting cell type (*Yamada et al., 2014*). We show that impairment of a global silencing regulator leads to reversion of neurogenesis-lineage committed cells to a less differentiated cell state, but also that this can be achieved by single ectopic expression of *kni*. This raises the possibility that embryonic TFs such as *kni* drive the establishment of a regulatory circuit that blocks differentiation. Although the involved factors of such mechanisms remain to be identified, we consider the identification of *kni* as a strong oncogene a valuable starting point for future studies.

The identification of a tumorigenic role of the embryonic TF Kni in *Drosophila*, is in line with the identification of other embryonic TFs playing a role in several different tumor models. For instance, aberrant expression of the embryonic TF Oct-4 blocks progenitor-cell differentiation and causes dysplasia in mouse adult epithelial tissues (*Kumar et al., 2012*; *Hochedlinger et al., 2005*). In humans, activation of the TF TAL1, normally expressed early in the erythroid lineage, has been shown to alter a core transcriptional regulatory circuit that in turn leads to tumor onset (T cell leukemia) (*Bradner et al., 2017*). Additionally, other relevant embryonic TFs, such as FOXF1, normally expressed in mesenchyme-derived cells, activate MAPK signaling when expressed in prostate epithelial cells and contribute to tumorigenesis (*Fulford et al., 2016*).

Taken together, our data show that *knirps* can drive tumor onset and is a strong oncogene in $ph^{505}$-tumors. Moreover, our work is consistent with a growing understanding between the connections of developmental gene expression and cancer. We hope that in the long-term these findings can contribute to the development of new therapies for cancers driven by misexpression of TFs.

## Inability to differentiate as target for therapy in embryonic-like tumors

Loss of differentiation capabilities, as well as the emergence of a progenitor-like state that promotes cellular transformation and tumor initiation are common processes observed in cancer (*Roy and Hebrok, 2015*; *Bossuyt et al., 2009*). The concept of dedifferentiation preceding tumorigenesis has been shown in *Drosophila* neurons, where neurons lacking the TF Lola dedifferentiate, turning on neural stem cell genes, begin to divide, and form tumors (*Southall et al., 2014*).

We provide evidence that kni-ectopic tumors, very similar to $ph^{505}$-tumors, also fail to undergo differentiation. Besides commonalities such as loss of polarity and loss of cell identity, these two tumor models also share the ectopic activation of the JAK/STAT signaling pathway. Developmental studies suggest the cooperation between JAK/STAT and gap genes (e.g. *knirps*) in regulating expression of pair-rule genes for segmentation during embryogenesis (*Hou et al., 2002*). Hyperactivation of the JAK/STAT pathway has been observed in different human cancers, where it activates survival and proliferation genes (*Buchert et al., 2016*). Also, cells can be maintained in a less differentiated and more proliferative state by JAK/STAT pathway activation, as highlighted by its activation in stem cell niches in *Drosophila* (*Hou et al., 2002*; *Amoyel and Bach, 2012*; *Christofi and*

*Apidianakis, 2013*) and in mouse embryonic stem cells (*Hao et al., 2006*). Furthermore, there is evidence suggesting that this pathway must be switched off to allow differentiation of hematopoietic progenitors in flies (*Amoyel and Bach, 2012*). In agreement with this, blocking JAK/STAT activity suppresses the PRC1 mutant tumor phenotype (*Classen et al., 2009*) and in our hands induces the re-establishment of the differentiation program characteristic of eye-antennal imaginal discs.

However, using the blockage of signaling pathways as a therapeutic target has been shown to be difficult due to the redundancy of the signaling networks and thus acquired drug resistance is common in cancer cells (*Gonda and Ramsay, 2015*; *Buchert et al., 2016*). Alternatively, forced differentiation by means of TF activation might solve this issue. We used *atonal*, a pro-neural TF in eye discs (*Bossuyt et al., 2009*) and downregulated in $ph^{505}$-tumors, to ultimately restore differentiation in eye-antennal tissues. This approach proved to be sufficient to prevent tumor cells from proliferating, reduce tumor burden and recover the normal pattern of differentiation.

The significance of these observations, referred to as 'differentiation therapy', is supported by work done in acute myeloid leukemia where therapies to overcome the cellular differentiation arrest have led to favorable outcomes (*Gocek and Marcinkowska, 2011*). Moreover this strategy has also been suggested to restrict the cellular plasticity of cancer stem cells (*Wainwright and Scaffidi, 2017*). Our findings highlight the importance of embryonic transcription factors in oncogenesis and favor the potential of re-establishing differentiation as an attractive alternative in future considerations for cancer therapy.

## Materials and methods

### Key resources table

| Reagent type (species) or resource | Designation | Source or reference | Identifiers | Additional information |
|---|---|---|---|---|
| genetic reagent (D. melanogaster) | OregonR (host flies) | Bloomington Drosophila Stock Center | BL25211 | |
| genetic reagent (D. melanogaster) | w[1118] (host flies) | Bloomington Drosophila Stock Center | BL5905 | |
| genetic reagent (D. melanogaster) | yw, FRT19A | Bloomington Drosophila Stock Center | BL1744 | |
| genetic reagent (D. melanogaster) | FRT19A, ph505/FM7 act-GFP | A.-M. Martinez | ph505 | |
| genetic reagent (D. melanogaster) | w, tubGal80, FRT19A; eyFlp5, Act5C > y +> Gal4, UAS-GFP.S56T | T. Xu | 19A Tester | |
| genetic reagent (D. melanogaster) | P{w[+mC]=tubP-GAL80}LL1, w* P{ry[+t7.2]=ey FLP.N}2P {ry[+t7.2]=neoFRT}19A | Bloomington Drosophila Stock Center | BL42717 | |
| genetic reagent (D. melanogaster) | w[1118]; P{w[+mC]=GAL4-Act5C(FRT.CD2).P}S, P{w[+mC]=UAS RFP.W}3/TM3 | Bloomington Drosophila Stock Center | BL30558 | |
| genetic reagent (D. melanogaster) | w; UAS-domeΔCYT | J. Hombría | domeΔCYT | |
| genetic reagent (D. melanogaster) | UAS-ph L7 (#3) | F. Maschat | UAS-ph | |
| genetic reagent (D. melanogaster) | w; UAS-p35 | S. Kurata | UAS-p35 | |
| genetic reagent (D. melanogaster) | UAS-Kni/Cyo | M. Affolter | UAS-Kni | |
| genetic reagent (D. melanogaster) | UAS-ato (#2) | G. Mardon | UAS-ato | |
| genetic reagent (D. melanogaster) | w1118;; P{NRE-EGFP.S}1 | Bloomington Drosophila Stock Center | BL30728 (Notch reporter, NRE-GFP on #3) | |
| genetic reagent (D. melanogaster) | w;;TRE-DsRed | Chatterjee & Bohmann 2012 | JNK reporter | |

*Continued on next page*

*Continued*

| Reagent type (species) or resource | Designation | Source or reference | Identifiers | Additional information |
|---|---|---|---|---|
| genetic reagent (D. melanogaster) | w[1118]; P{w[+mC]=10XStat92E-GFP}2 | Bloomington Drosophila Stock Center | BL26198 (JAK/STAT reporter) | |
| genetic reagent (D. melanogaster) | RNAi of Kni: y(*Abate-Shen, 2002*) sc[*] v(*Abate-Shen, 2002*); P{y[+t7.7] v[+t1.8]=TRiP. HMS01184}attP2 | Bloomington Drosophila Stock Center | Trip BL34705 (kni-KD) | |
| genetic reagent (D. melanogaster) | RNAi of Kni: y(*Abate-Shen, 2002*) v(*Abate-Shen, 2002*); P{y[+t7.7] v[+t1.8]=TRiP.JF02544}attP2 | Bloomington Drosophila Stock Center | Trip BL27259 (kni-KD (2)) | |
| genetic reagent (D. melanogaster) | RNAi of Abd-A: y(*Abate-Shen, 2002*) v(*Abate-Shen, 2002*); P{y[+t7.7] v[+t1.8]=TRiP.JF03167}attP2 | Bloomington Drosophila Stock Center | Trip BL28739 (Abd-A-KD) | |
| genetic reagent (D. melanogaster) | RNAi of lms: y(*Abate-Shen, 2002*) sc[*] v(*Abate-Shen, 2002*); P{y[+t7.7] v[+t1.8]=TRiP.HMS02709}attP2/TM3, Sb(*Abate-Shen, 2002*) | Bloomington Drosophila Stock Center | Trip BL43995 (lms-KD) | |
| genetic reagent (D. melanogaster) | RNAi of gsc: y(*Abate-Shen, 2002*) sc[*] v(*Abate-Shen, 2002*); P{y[+t7.7] v [+t1.8]=TRiP.HMC02397}attP2 | Bloomington Drosophila Stock Center | Trip BL50894 (gsc-KD) | |
| genetic reagent (D. melanogaster) | RNAi of cad: y(*Abate-Shen, 2002*) sc[*] v(*Abate-Shen, 2002*); P{y[+t7.7] v[+t1.8]=TRiP.HMC04863}attP40 | Bloomington Drosophila Stock Center | Trip BL57546 (cad-KD) | |
| genetic reagent (D. melanogaster) | RNAi of bgcn: y(*Abate-Shen, 2002*) sc[*] v(*Abate-Shen, 2002*); P{y[+t7.7] v[+t1.8]=TRiP .GL00596}attP40 | Bloomington Drosophila Stock Center | Trip BL36636 (bgcn-KD) | |
| genetic reagent (D. melanogaster) | RNAi of Sox100b: y(*Abate-Shen, 2002*) sc[*] v(*Abate-Shen, 2002*); P{y[+t7.7] v[+t1.8]=TRiP.GLV21021}attP2 | Bloomington Drosophila Stock Center | Trip BL35656 (Sox100b-KD) | |
| genetic reagent (D. melanogaster) | RNAi of btd: y(*Abate-Shen, 2002*) v(*Abate-Shen, 2002*); P{y[+t7.7] v[+t1.8]=TRiP.JF03389}attP2 | Bloomington Drosophila Stock Center | Trip BL29453 (btd-KD) | |
| genetic reagent (D. melanogaster) | RNAi of eve: y(*Abate-Shen, 2002*) v(*Abate-Shen, 2002*); P{y[+t7.7] v[+t1.8]=TRiP.JF03161}attP2 | Bloomington Drosophila Stock Center | Trip BL28734 (eve-KD) | |
| genetic reagent (D. melanogaster) | RNAi of tin: y(*Abate-Shen, 2002*) v(*Abate-Shen, 2002*); P{y[+t7.7] v[+t1.8]=TRiP.HMC03064}attP2 | Bloomington Drosophila Stock Center | Trip BL50663 (tin-KD) | |
| genetic reagent (D. melanogaster) | RNAi of Dr: y(*Abate-Shen, 2002*) v(*Abate-Shen, 2002*); P{y[+t7.7] v[+t1.8]=TRiP.HMC03402}attP2 | Bloomington Drosophila Stock Center | Trip BL51830 (Dr-KD) | |
| genetic reagent (D. melanogaster) | RNAi of nub: y(*Abate-Shen, 2002*) v(*Abate-Shen, 2002*); P{y[+t7.7] v[+t1.8]=TRiP.HMC03992}attP2 | Bloomington Drosophila Stock Center | Trip BL55305 (nub-KD) | |
| genetic reagent (D. melanogaster) | RNAi of fkh: y(*Abate-Shen, 2002*) sc[*] v(*Abate-Shen, 2002*); P{y[+t7.7] v[+t1.8]=TRiP.HMS01103}attP2 | Bloomington Drosophila Stock Center | Trip BL33760 (fkh-KD) | |
| genetic reagent (D. melanogaster) | RNAi of Abd-B: y(*Abate-Shen, 2002*) v(*Abate-Shen, 2002*); P{y[+t7.7] v[+t1.8]=TRiP.JF02309}attP2 | Bloomington Drosophila Stock Center | Trip BL26746 (Abd-B-KD) | |
| genetic reagent (D. melanogaster) | RNAi of pb: y(*Abate-Shen, 2002*) v(*Abate-Shen, 2002*); P{y[+t7.7] v[+t1.8]=TRiP.HMC03065}attP2 | Bloomington Drosophila Stock Center | Trip BL50664 (pb-KD) | |
| genetic reagent (D. melanogaster) | RNAi of grn: y(*Abate-Shen, 2002*) sc[*] v(*Abate-Shen, 2002*); P{y[+t7.7 ] v[+t1.8]=TRiP.HMS01085}attP2 | Bloomington Drosophila Stock Center | Trip BL33746 (grn-KD) | |

*Continued on next page*

*Continued*

| Reagent type (species) or resource | Designation | Source or reference | Identifiers | Additional information |
|---|---|---|---|---|
| genetic reagent (D. melanogaster) | RNAi of odd: y(*Abate-Shen, 2002*) sc[*] v(*Abate-Shen, 2002*); P{y[+t7.7] v[+t1.8]=TRiP.HMS01315}attP2/TM3, Sb(*Abate-Shen, 2002*) | Bloomington Drosophila Stock Center | Trip BL34328 (odd-KD) | |
| genetic reagent (D. melanogaster) | RNAi of croc: y(*Abate-Shen, 2002*) sc[*] v(*Abate-Shen, 2002*); P{y[+t7.7] v[+t1.8]=TRiP.HMS01122}attP2 | Bloomington Drosophila Stock Center | Trip BL34647 (croc-KD) | |
| genetic reagent (D. melanogaster) | RNAi of drm: y(*Abate-Shen, 2002*) v(*Abate-Shen, 2002*); P{y[+t7.7] v[+t1.8]=TRiP.HMJ02120}attP40 | Bloomington Drosophila Stock Center | Trip BL42548 (drm-KD) | |
| genetic reagent (D. melanogaster) | RNAi of Doc2: y(*Abate-Shen, 2002*) sc[*] v(*Abate-Shen, 2002*); P{y[+t7.7] v[+t1.8]=TRiP.HMS02804}attP2 | Bloomington Drosophila Stock Center | Trip BL44087 (Doc2-KD) | |
| genetic reagent (D. melanogaster) | RNAi of Doc3: y(*Abate-Shen, 2002*) v(*Abate-Shen, 2002*); P{y[+t7.7] v[+t1.8]=TRiP.JF02223}attP2 | Bloomington Drosophila Stock Center | Trip BL31932 (Doc3-KD) | |
| genetic reagent (D. melanogaster) | RNAi of vvl: y(*Abate-Shen, 2002*) v(*Abate-Shen, 2002*); P{y[+t7.7] v[+t1.8]=TRiP.JF02126}attP2 | Bloomington Drosophila Stock Center | Trip BL26228 (vvl-KD) | |
| genetic reagent (D. melanogaster) | RNAi of Doc1: y(*Abate-Shen, 2002*) v(*Abate-Shen, 2002*); P{y[+t7.7] v[+t1.8]=TRiP.JF02222}attP2 | Bloomington Drosophila Stock Center | Trip BL31931 (Doc1-KD) | |
| genetic reagent (D. melanogaster) | RNAi of Kr: y(*Abate-Shen, 2002*) v(*Abate-Shen, 2002*); P{y[+t7.7] v[+t1.8]=TRiP.JF02745}attP2 | Bloomington Drosophila Stock Center | Trip BL27666 (Kr-KD) | |
| antibody | Ph (rabbit) | R. Paro Lab | N/A | (1:100) |
| antibody | Arm (mouse) | DSHB | N27A1 | (1:5) |
| antibody | MMP1 (mouse) | DSHB | 5H7B11 | (1:300) |
| antibody | ELAV (rat) | DSHB | 7E8A10 | (1:30) |
| antibody | Eya (mouse) | DSHB | eya10H6 | (1:500) |
| antibody | Hth (goat) | H. Sun | dG20, Santa Cruz | (1:100) |
| antibody | Eve (mouse) | DSHB | Eve 3C10 | (1:100) |
| antibody | Cad (rabbit) | P. Macdonald Lab | #1 | (1:500) |
| antibody | Abd-B (mouse) | DSHB | N/A | (1:10) |
| antibody | Dcp-1 (rabbit) | Cell Signaling | 9578S | (1:200) |
| antibody | Phalloidin Alexa 633 | Life Technologies | A22284 | (1:400) |
| antibody | phospho-Histone H3, Ser 10 (pH3, rabbit) | Millipore | 06–570 | (1:200) |
| antibody | Alexa 568- or 594 secondaries | Life Technologies | A-11036, A11031, A-11077, A-11058 | (1:500) |
| commercial assay or kit | PicoPure RNA Isolation Kit | Thermo Fisher | KIT0204 | |
| commercial assay or kit | RNase-Free DNase Set | Qiagen | #79254 | |
| commercial assay or kit | Quant-iT RiboGreen RNA Assay Kit | Thermo Fisher | R11490 | |
| chemical compound, drug | Collagenase | Sigma | C1639 | |
| software, algorithm | iRegulon | *Janky et al. (2014)* | http://iregulon.aertslab.org | |

*Continued on next page*

*Continued*

| Reagent type (species) or resource | Designation | Source or reference | Identifiers | Additional information |
|---|---|---|---|---|
| software, algorithm | Trimmomatic | *Bolger et al. (2014)* | http://www.usadellab.org/cms/?page=trimmomatic | |
| software, algorithm | FastQC | FastQC A Quality Control tool for High Throughput Sequence Data (v0.11.2) | www.bioinformatics.babraham.ac.uk/projects/fastqc/ | |
| software, algorithm | STAR | *Dobin et al. (2013)* | https://github.com/alexdobin/STAR | |
| software, algorithm | Picard Tools | Picard tools (version 1.121) | www.broadinstitute.github.io/picard/ | |
| software, algorithm | HTSeq | *Anders et al. (2015)* | http://www-huber.embl.de/HTSeq/doc/overview.html# | |
| software, algorithm | DESeq2 | *Love et al. (2014)* | https://bioconductor.org/packages/release/bioc/html/DESeq2.html | |
| software, algorithm | WEB-based GEne SeT AnaLysis Toolkit | *Wang et al., 2013* | WebGestalt: www.webgestalt.org | |
| software, algorithm | Ilastik | *Sommer, 2011* | http://ilastik.org | |
| software, algorithm | Matlab | MATLAB 2016b, The Math Works Inc., Natick, MA | https://ch.mathworks.com/products/matlab.html | |
| software, algorithm | Imaris | Imaris v 8.4.1 (Build 41809 for x64), Bitplane AG | http://www.bitplane.com/imaris/imaris | |
| software, algorithm | GraphPad Prism 7.0 | GraphPad Prism version 7.00 for Windows, GraphPad Software, La Jolla California USA | https://www.graphpad.com/scientific-software/prism/ | |

## Contact for reagent and resource sharing

Further information and requests for resources and reagents should be directed to and will be fulfilled by the corresponding authors.

## Experimental model

Flies were maintained on standard food at 25°C and 60% relative humidity, under a 12 hr light: 12 hr dark cycle. All fly stocks used are listed in the Key Resources Table.

## Mitotic recombination and generation of clones

Mitotic recombination was induced by the expression of FLP recombinase under the control of eyeless promoter (eyFlp). Additionally, using the mosaic analysis with a repressible cell marker (MARCM) system (*Wu and Luo, 2006*), clones were fluorescently labeled with GFP. For our mutant experiments, we used $ph^{505}$ allele to knock-out both genes in the $ph$ locus ($ph$-$p$ and $ph$-$d$). For control experiments, MARCM clones were generated with a FRT19A blank stock line. Specifically, '19A tester' stock line was crossed either with FRT19A, $ph^{505}$/FM7 act-GFP or with FRT19A in order to generate mutant or control clones in eye-antennal imaginal discs, respectively. Larvae were examined at the late third instar stage.

## Candidate hit-validation in vivo – RNAi constructs

RNAi strains were initially balanced (#2, Cyo or #3 TM6b) and subsequently crossed with the strain carrying the mutant allele and maintained as a stock. For generation of clones and simultaneous expression of RNAi-target, the stock mentioned above was crossed with '19A tester' strain. For all final crosses 25 female virgins were crossed with eight males, in order to insure that number of larvae per fly food vial would be similar and not overcrowded. Two independent crosses for each RNAi were performed. Up to three replicates were collected from each RNAi cross.

Confirmation of the results obtained by RNAi KD with a *knirps^mut* allele could not be realized. We did not succeed in generating a recombinant mutant allele (Kni[FC13]) with a FRT element, probably caused by the expected low frequency of recombination between the two elements.

For determination of eclosion rates, larvae were selected accordingly to GFP expression in eye discs, counted and transferred to a new food vial. After eclosion the number of adults was counted. Eclosion rate was measured as the ratio of number larvae over the number of adults that hatched. Images of adult eyes were acquired with Nikon SMZ1270.

## Immunostaining of eye-antennal imaginal discs

Third instar larvae were dissected in PBS 1x and fixed in 4% paraformaldehyde (SIGMA, #P6148) in PBS 1x for 20 min at room temperature (RT) and washed with PBS with 0.1% TritonX-100 (SIGMA, #T9284) (0.1% PBS-T) for 30 min (3 × 10 min) and blocked (0.1% Bovine Serum Albumin (Serva, #11930.04) in 0.1% PBS-T) for 1 hr at RT. Larvae were then incubated with primary antibodies in blocking solution overnight at 4°C, washed with 0.1% PBS-T (3 × 15 min) and incubated with secondary antibodies in blocking solution for 2 hr at RT. After washing with 0.1% PBS-T for 15 min, DAPI (Invitrogen #62248, 1:500) was added and incubated for 15 min at RT. Imaginal discs were then dissected in PBS 1x and mounted in a slide with Vectashield mounting medium (Vector Laboratories).

The primary antibodies used in this study were: rabbit anti-Ph (Paro lab; 1:100), mouse anti-Arm (DSHB N27A1; 1:5), mouse anti-MMP1 (DSHB 5H7B11; 1:300), rat anti-ELAV (DSHB 7E8A10; 1:30), mouse anti-eve (DSHB Eve3C10; 1:100), rabbit anti-cad (Macdonald lab; 1:500), mouse anti-Eya (DSHB eya10H6; 1:500), goat anti-Hth (H. Sun; 1:100), mouse anti-Abd-B (DSHB; 1:10), rabbit anti-Dcp-1 (Cell Signaling 9578S; 1:200), rabbit anti-pH3 (Millipore 06–570; 1:200).

Appropriate combinations of Alexa-coupled secondary antibodies were subsequently applied. Phalloidin-633 (Life Technologies A22284, 1:100) was used for actin staining. The secondary antibodies used were: goat anti-Rabbit Alexa 568 (Life Technologies., Bleiswijk, Netherlands, A-11036), goat anti-mouse Alexa 568 (Life Technologies, A-11031), goat anti-rat Alexa 568 (Life Technologies, A-11077), donkey anti-goat Alexa 594 (Life Technologies, A-11058). All secondary antibodies were used at 1:500 dilutions.

Samples were analyzed with a Leica SP5 or SP8 confocal microscope. Images were processed using ImageJ and were assembled with Adobe Photoshop.

## Transplantations

Transplantation assays were performed according to previous reports (*Rossi and Gonzalez, 2015*). Briefly, eye-antennal discs of genotypes of interest (either $ph^{505}$; $ph^{505}$, kni-KD; or FRT19A, UAS-Kni) were cut into small pieces and transplanted into the abdomen of female hosts (w[1118] or wild-type Oregon R). Transplanted hosts were kept at 25°C and monitored for GFP$^+$ overgrowth mass. Number of tumor-bearing hosts was assessed every week upon transplantation. Transplanted hosts with $ph^{505}$ tissues were used as control to account for pathogen contaminations, temperature changes or other issues that could affect the survival of the flies. Adult hosts were analyzed and images were acquired with Nikon SMZ1270.

## Workflow for sample preparation for RNA-sequencing

Protocol for sample preparation for RNA-sequencing was adapted from published work (*Harzer et al., 2013*; *Martinez et al., 2009*; *Dutta et al., 2013*). Each biological replicate for FACS was composed of a total of 200–250 eye-antennal imaginal discs of third instar larvae dissected in PBS 1x. After spinning down and removing PBS 1x, imaginal discs were placed in low-binding 1.5 mL tube with 200 uL of saline solution containing collagenase (25 discs/tube) (collagenase SIGMA, C1639 - 1.5 mg/mL diluted in Rinaldini's saline solution) and incubated at RT for 45 min, 300rpms. Tubes were agitated every 15 min and mechanical digestion was performed twice during collagenase incubation (pipetting up-and-down with 27G syringe). After digestion, tubes were pooled in a total of 2 1.5 mL tubes and centrifuged for 25 min, 300 g, 4°C. Supernatant was removed and pellet was resuspended in PBS 1x. Solution was filtered and shortly kept on ice before proceeding for FACS. Several rounds of FACS-sorting were performed from pools of $ph^{505}$ eye-antennal discs, using a BD FACS Aria cell sorter (BD Biosciences) of the FMI FACS facility (FMI, Basel) and data was collected on the basis of FSC/SSC parameters. Sorting time was kept below 45 min to insure the maximum viability of the cells. Two populations of cells were collected separately, GFP$^+$- (mutant cells) and GFP$^-$-sorted cells (control), directly into extraction buffer (200 μL, PicoPure RNA isolation kit, Thermo Fisher, KIT0204). RNA extraction was performed accordingly to manufacturers instructions,

including a step of DNase treatment (Qiagen, catalog #79254). Samples were eluted in the final volume of 11 µL and kept at −80°C. RNA concentration (RiboGreen, ThermoFisher, #R11490) and integrity (Fragment analyzer, AATI) of sorted samples was assessed by the Genomics Facility Basel (D-BSSE, Basel). From the several rounds of samples' preparation, we choose 4 pairs of samples (tumor and matched-control) and three extra tumor samples from batches where control cells did not have the desired quality, to prepare libraries for sequencing. Due to the low amount of RNA in these samples, libraries were prepared by the Genomics Facility Basel using a method conceived for single cell RNA-seq (Smart-seq2) (*Picelli et al., 2014*).

## RNA-sequencing – Differential expression analysis

The following steps were performed on 22 libraries. There were two technical replicates *per* sample corresponding to a total of 11 samples (seven tumor, four control). The libraries were sequenced in paired-end mode (2 × 150 bp) in a NextSeq500 (Illumina), and insert sizes around 300 bp (ungapped forward and reverse tags). Adaptor clipping and quality trimming was performed with Trimmomatic (*Bolger et al., 2014*) (v0.30), after initial quality checks with FastQC (v0.11.2, *www.bioinformatics. babraham.ac.uk/projects/fastqc/*). Reads were aligned using the splice aware aligner STAR (*Dobin et al., 2013*) (v2.3.0e) and subsequently filtered to remove potential PCR-duplicates with Picard Tools (v1.121, *broadinstitute.github.io/picard/*). Transcript counts were produced with HTSeq (*Anders et al., 2015*) (v0.6.1) using the Ensembl 78 annotation (*Aken et al., 2016*). The subsequent differential expression analysis was performed in R (v3.1.0, *www.r-project.org*) using the DESeq2 package (*Love et al., 2014*) (v1.6.1), neglecting one library (technical replicate), which did not meet quality standards. All the differentially expressed genes were submitted to the"WEB-based GEne SeT AnaLysis Toolkit' (WebGestalt (*Wang et al., 2013*), *www.webgestalt.org*), submitting either all differentially expressed genes at the same time, or splitting them into up- and down-regulated genes.

For the in vivo screen, we decided not to exclude candidates based on their log2 fold change, as is commonly done, but rather selected candidates based on a stringent adjusted p value (padj. <0.01) and a-priori knowledge.

## Incorporation of other gene expression datasets

RNA-seq profiles of our tumor and control samples were compared with available *D. melanogaster* datasets (*Figure 1—source data 3*), specifically comparing 124 differentially expressed TF-encoding genes (*Figure 1—source data 4*). All additional samples (fastq-files) were obtained from the Gene Expression Omnibus (GEO, *www.ncbi.nlm.nih.gov/geo/*) and processed in a similar fashion as the original 11 samples. For single-end-libraries, the removal of duplicates was not performed. Settings in Trimmomatic were adjusted for each sample, taking into account the sequencer type and read lengths. All samples were aligned with STAR and counting was performed with HTSeq.

## Hierarchical clustering

Hierarchical clustering was performed after normalizing gene-expression values with DESeq2. The expression values after variance stabilizing transformation were then mean-centered for each gene. Hierarchical clustering was performed between samples using 1-Pearson correlation as distance measure, while genes were clustered using Euclidean distance. The datasets used for comparison were retrieved from the following references: (*Graveley et al., 2011*; *Gan et al., 2010*; *Jüschke et al., 2013*; *Potier et al., 2014*; *Berger et al., 2012*; *Naval-Sánchez et al., 2013*; *Czech et al., 2013*; *Atkins et al., 2016*).

## Image analysis

Images of eye-antennal imaginal tissues were acquired using 20x or 40x objectives on the Leica SP5/SP8 confocal microscopes and processed using ImageJ or Imaris. Images of adult eyes or transplantation hosts were acquired with Nikon SMZ1270.

## Quantification pipeline for tumor volumes

As a measure of tumor volume, we quantified the space taken up by the tumor in these tissues employing a quantification pipeline developed in our lab (*Beira et al., 2018*). To automate image

segmentation and identification of clones across imaginal discs, we used Ilastik (Interactive Learning and Segmentation Toolkit, [*Sommer, 2011*]) to build an unbiased supervised learning classification of clone regions and surrounding tissue (with 5 $ph^{505}$-tumor eye-antennal imaginal discs). Confocal images of tissues of interest were acquired with a 0.8–1.1 µm z-stacks. The classification method was then used for the test set of $ph^{505}$-tumor tissues (N = 50), as well as upon perturbation (either TF-KD or overexpression of *ph*, p35 and *ato*). After unbiased classification of clones, a Matlab script (kindly developed by Aaron Ponti, SCF, D-BSSE) was used to enable us to use Imaris (Bitplane) in order to obtain volume data for each spatially defined clone, total clone number *per* tissue, and tissue volume (DAPI). Tumor volume (%) was then calculated as the ratio of tumor volume (sum up volumes of all GFP-clones in a tissue) over the size of the respective tissue (volume, DAPI).

## Quantification of phospho-histone H3 cells

In order to measure proliferation levels, we quantified the number of phospho-histone H3 (pH3) positive cells within eye-antennal imaginal discs in the four conditions of interest (FRT19A; FRT19A, UAS-ato; $ph^{505}$; $ph^{505}$, UAS-ato). We used Imaris (Bitplane) for semi-automated image segmentation of total tissue volume (DAPI), total volume of clones (GFP+ cells) and number of pH3+ cells. In addition, we used the segmented GFP signal to mask voxels of the pH3 +channel inside and outside of GFP positive cells to zero. In this way we were able to measure pH3+ cells inside and outside the clones. To account for differences in the size of tissues and clones, we normalized the data accordingly. For the 'whole tissue' condition, total numbers of pH3+ cells were normalized to total tissue volume (*per* tissue); for 'inside clones', numbers of pH3+ cells within clones were normalized to volume of GFP+ cells *per* tissue; for 'outside clones', numbers of pH3+ cells outside of GFP+ clones were normalized to volume of GFP- cells *per* tissue. Values of pH3+ cells are represented *per* mm$^3$.

## Statistical analysis

GraphPad Prism 7.0 was used for statistical analysis and generation of the graphical output. No statistical analysis was used to predetermine sample size. Sample sizes (N) and p-values are indicated in the figures and/or figure legends. Statistical tests used: Kruskal-Wallis with Dunn's multiple comparisons test for eclosion rate, tumor volume (%), number of clones and number of pH3+ cells; one-way ANOVA with Dunnett's multiple comparisons test for tissue size and average tumor volume. ****p<0.0001; ***p<0.001; **p<0.01; *p<0.05. All data points represented by dots in the plots for tumor volume, average tumor volume, tissue volume, number of clones and number of pH3+ cells *per* tissue are randomly distributed along x-axis.

## Data availability

The accession number for the sequencing data reported in this paper is GEO: GSE101463.

## Acknowledgements

The authors would like to thank Hubertus Kohler from FMI for support with FACS sorting, D-BSSE Single Cell Facility for support with confocal microscopy and Aaron Ponti for help with the image quantification pipeline, D-BSSE Genomics Facility Basel for support with sequencing and Katja Eschbach for library preparation. We thank H Sun, P Macdonald, A-M Martinez, T Xu, J Hombría, F Maschat, M Affolter, S Kurata and G Mardon for sharing reagents and fly stocks. Fly stocks from the Bloomington Drosophila Stock Center (NIH P40OD018537) and antibodies from Developmental Studies Hybridoma Bank (DSHB) were also used in this study. JT is a member of the Life Science Zürich Graduate School (PhD Program in Molecular Life Sciences). JVB is supported by an EMBO long-term fellowship (ALTF1131-2014 and Marie Curie Actions LTFCOFUND2013/GA-2013–609409). Research in RP's laboratory is funded by the Swiss National Science Foundation (SNF grant 31003A_143922) and the ETH Zürich.

# Additional information

## Funding

| Funder | Grant reference number | Author |
|---|---|---|
| Schweizerischer Nationalfonds zur Förderung der Wissenschaftlichen Forschung | 31003A_143922 | Renato Paro |
| ETH Zurich | | Joana Torres<br>Remo Monti<br>Ariane L Moore<br>Makiko Seimiya<br>Yanrui Jiang<br>Niko Beerenwinkel<br>Christian Beisel<br>Renato Paro |
| European Molecular Biology Organization | ALTF1131-2014 | Jorge V Beira |

The funders had no role in study design, data collection and interpretation, or the decision to submit the work for publication.

## Author contributions

Joana Torres, Conceptualization, Investigation, Methodology, Writing—original draft; Remo Monti, Formal analysis, Writing—review and editing; Ariane L Moore, Formal analysis; Makiko Seimiya, Yanrui Jiang, Investigation; Niko Beerenwinkel, Christian Beisel, Resources, Funding acquisition; Jorge V Beira, Investigation, Methodology, Supervision, Writing—review and editing; Renato Paro, Conceptualization, Resources, Supervision, Funding acquisition, Methodology, Writing—review and editing

## Author ORCIDs

Joana Torres http://orcid.org/0000-0002-5651-4575
Niko Beerenwinkel http://orcid.org/0000-0002-0573-6119
Jorge V Beira http://orcid.org/0000-0002-2884-4964
Renato Paro http://orcid.org/0000-0003-3308-2965

## Decision letter and Author response

Decision letter https://doi.org/10.7554/eLife.32697.051
Author response https://doi.org/10.7554/eLife.32697.052

# Additional files

## Supplementary files

• Transparent reporting form
DOI: https://doi.org/10.7554/eLife.32697.024

## Major datasets

The following dataset was generated:

| Author(s) | Year | Dataset title | Dataset URL | Database, license, and accessibility information |
|---|---|---|---|---|
| Torres J, Monti R, Hofmann AL, Seimiya M, Jiang Y, Beerenwinkel N, Beisel C, Beira JV, Paro R | 2017 | An epigenetically-deregulated tumor state reveals a new role for embryonic transcription factors as cancer drivers | https://www.ncbi.nlm.nih.gov/geo/query/acc.cgi?acc=GSE101463 | Publicly available at the NCBI Gene Expression Omnibus (accession no: GSE101463) |

The following previously published datasets were used:

| Author(s) | Year | Dataset title | Dataset URL | Database, license, and accessibility information |
|---|---|---|---|---|
| Czech B, Preall JB, McGinn J, Hannon G | 2013 | A transcriptome-wide RNAi screen in the Drosophila ovary reveals factors of the germline piRNA pathway | https://www.ncbi.nlm.nih.gov/geo/query/acc.cgi?acc=GSE46100 | Publicly available at the NCBI Gene Expression Omnibus (accession no: GSE46100) |
| Naval-Sánchez M, Potier D, Haagen L, Sánchez M, Munck S, Van de Sande B, Casares F, Christiaens V, Aerts S | 2013 | RNA-seq in wild-type and glass mutant eye-antennal discs in Drosophila melanogaster | https://www.ncbi.nlm.nih.gov/geo/query/acc.cgi?acc=GSE39781 | Publicly available at the NCBI Gene Expression Omnibus (accession no: GSE39781) |
| Berger C, Harzer H, Burkard TR, Steinmann J, van der Horst S, Laurenson AS, Novatchkova M, Reichert H, Knoblich JA | 2012 | Transcriptome analysis of Drosophila neural stem cells reveals a transcriptional network for self-renewal. | https://www.ncbi.nlm.nih.gov/geo/query/acc.cgi?acc=GSE38764 | Publicly available at the NCBI Gene Expression Omnibus (accession no: GSE38764) |
| Potier D, Davie K, Hulselmans G, Naval Sanchez M, Haagen L, Huynh-Thu VA, Koldere D, Celik A, Geurts P, Christiaens V, Aerts S | 2014 | Mapping gene regulatory networks in Drosophila eye development by large-scale transcriptome perturbations and motif inference | https://www.ncbi.nlm.nih.gov/geo/query/acc.cgi?acc=GSE59059 | Publicly available at the NCBI Gene Expression Omnibus (accession no: GSE59059) |
| Jüschke C, Dohnal I, Pichler P, Harzer H, Swart R, Ammerer G, Mechtler K, Knoblich JA | 2013 | Transcriptome and proteome quantification of a tumor model provides novel insights into post-transcriptional gene regulation | https://www.ncbi.nlm.nih.gov/geo/query/acc.cgi?acc=GSE51412 | Publicly available at the NCBI Gene Expression Omnibus (accession no: GSE51412) |
| Gan Q, Chepelev I, Wei G, Tarayrah L, Cui K, Zhao K, Chen X | 2010 | Dynamic regulation of alternative splicing and chromatin structure in Drosophila gonads revealed by RNA-seq | https://www.ncbi.nlm.nih.gov/geo/query/acc.cgi?acc=GSE16960 | Publicly available at the NCBI Gene Expression Omnibus (accession no: GSE16960) |
| Negre N, Bild NA, White KP | 2009 | Genome-wide transcriptome sequencing at different stages of Drosophila development, RNA-seq | https://www.ncbi.nlm.nih.gov/geo/query/acc.cgi?acc=GSE18068 | Publicly available at the NCBI Gene Expression Omnibus (accession no: GSE18068) |
| Atkins M, Potier D, Romanelli L, Jacobs J, Mach J, Hamaratoglu F, Aerts S, Halder G | 2016 | An Ectopic Network of Transcription Factors Regulated by Hippo Signaling Drives Growth and Invasion of a Malignant Tumor Model. | https://www.ncbi.nlm.nih.gov/geo/query/acc.cgi?acc=GSE71350 | Publicly available at the NCBI Gene Expression Omnibus (accession no: GSE71350) |

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
