## [Decision Letter]

Thank you for submitting your article "A switch in transcription and cell fate governs the onset of an epigenetically-deregulated tumor in *Drosophila*" for consideration by *eLife*. Your article has been reviewed by three peer reviewers, and the evaluation has been overseen by a Reviewing Editor and K VijayRaghavan as the Senior Editor. The following individual involved in review of your submission has agreed to reveal his identity: Georg Halder (Reviewer #2).

The reviewers have discussed the reviews with one another and the Reviewing Editor has drafted this decision to help you prepare a revised submission.

Summary:

In this study, the authors introduce a tumor in *Drosophila* imaginal discs via loss of the epigenetic regulator polyhomeotic. They provide data to suggest that the acquisition of a tumor fate is due to a reprogramming event toward an earlier developmental stage. They further identify knirps as a key regulator of this new transcriptional program, providing the first evidence that Knirps can function as an oncogene. This interesting and well written paper contributes to an emerging theme in tumor development that tumors represent a distinct, but abnormal cell fate and that tumors have their own developmental networks. Torres, et al. utilize a tumor that is independent of Ras activation and find that the polyhomeotic tumors activate a different kind of program from that of previously characterized Ras driven tumors, bringing novelty to the field. Despite the strong difference in gene expression and network architecture, the authors observe similar roles for JNK and JAK/STAT signaling pathways in promoting many of the tumor phenotypes as has been noted for other tumors. Technically, the authors do a nice job of combining genetic and transcriptomic data. They also are successful with FACS and tumor transplant analyses, which are technically challenging in this system. However there were several points that all reviewers agreed must be addressed before this manuscript is acceptable for publication in *eLife*.

Essential revisions:

1) The authors posit that *ph^505^* cells may undergo dedifferentiation. This would be an interesting finding, but it would require first showing that the clones differentiate at an earlier developmental time point, then lose that cell fate. If the authors truly believe that dedifferentiation occurs they should show data which supports this conclusion. As the *ph* clones are induced in first instar, and transcriptomics supports that they have an embryo-like fate, it may be that the clones simply fail to undergo differentiation. The authors must show data to support this claim, or re-word the manuscript. For example, what is the transcriptome of a first instar discs?

2) The control of the RNA-seq are the GFP-negative cells from FACS sorting. What about differential expression compared to wild type EA discs?

3) Among the 75 samples used to compare, why was Ras/Scrib not included? That is typically used and for this tumour plenty of microarray and RNA-seq data are available.

4) How general are their findings? Do *ph* clones give rise to tumors in other tissues as well? The authors should test the effects of *ph* clones in other tissues, and address autonomy of *ph* clones effects.

5) Autonomous vs. non-autonomous effects should be addressed: ELAV that marks photoreceptors (PR) is used as a differentiation marker. *ph* clones lose ELAV and it is re-established when *kni*, the embryonic TF, is knocked down or Ato, required for PR differentiation, is expressed in these cells. What is puzzling is that adult eyes from discs with *ph* tumors display normal looking PRs in higher numbers. If *ph* mutant cells themselves are ELAV negative, then the surrounding wild-type cells must be forming these excessive PRs. As the authors cite (Feng et al., 2011), there is a non-autonomous component to the *ph* phenotype, which was proposed to be caused by JAK/STAT induction in neighbouring cells. So, are the tumors formed by the *ph* cells or their wt neighbours? This is a very important point as it changes the way the RNA-seq data would be interpreted. The authors state in the Discussion that the differential expression analysis was done comparing GFP (+) and (-) cells in discs with *ph* mutants. I think, comparing GFP (+) cells from discs with *ph* mutants to GFP (+) cells from discs with wt clones would be important as well, the same goes for GFP (-) cells.

6) Is Knirps a key regulator of the downregulated genes? The authors speculate on this point, but should include the analysis of the downregulated genes in iRegulon as this would reveal if Knirps is implicated as a key regulator in both sets.

7) Ato overexpressing clones in normal tissue appear to be small. The ELAV pattern in these discs is also somewhat disrupted. While overexpression of Ato clearly prevents tumor overgrowth in this model, it is unclear if this is due to actually inducing differentiation or if Ato overexpression simply prevents cells from proliferating. The authors should clarify this point, which may require EdU or pH3 stainings to address the proliferation, along with quantifications of normal relative to tumor clones sizes.

---

## [Author Response]

Essential revisions:1) The authors posit that *ph^505^* cells may undergo dedifferentiation. This would be an interesting finding, but it would require first showing that the clones differentiate at an earlier developmental time point, then lose that cell fate. If the authors truly believe that dedifferentiation occurs they should show data which supports this conclusion. As the ph clones are induced in first instar, and transcriptomics supports that they have an embryo-like fate, it may be that the clones simply fail to undergo differentiation. The authors must show data to support this claim, or re-word the manuscript. For example, what is the transcriptome of a first instar discs?

We thank reviewers for bringing up this issue to our attention. Indeed, the term ‘dedifferentiation’ was used to refer to a developmental reversion or cellular reprogramming event, such as eye-antennal imaginal disc to embryo-like identity. Ph-tumors do not express neuronal markers and express embryonic TFs, suggestive of a switch of cell fate, potentially through a dedifferentiation step. Dedifferentiation is a common event observed in cancer development (Southall et al., 2014). However, we agree that if the term dedifferentiation is used in the context of eye-antennal disc development (as pointed out by the reviewers), only tumors arising from differentiated cells (late third instar) would in principle be able to go through dedifferentiation, as first instar larvae discs are composed of undifferentiated cells.

In the system we used, MARCM-GFP clones are induced from an earlier stage of eye-antennal disc development. We confirmed this again ourselves by checking GFP-signals in late first-instar discs (Author response image 1). Technically, the dissection of first-instar imaginal discs is very challenging. We succeeded to isolate imaginal discs attached to brain tissue for imaging, but we do not have the technical capabilities required for dissection of first-instar discs that are intact and in a considerable amount required for transcriptomic analysis (several hundreds). We also could not find published gene expression datasets from first-instar EA discs (to our knowledge, these are not publicly available).

**Author response image 1. respfig1:** Eye-antennal imaginal discs from different developmental stages carrying *ph*^*505*^-clones (**A**-**C**). Mutant cells are GFP labeled and present in late first instar EA discs (**A**). Pictures were taken with the same objective in a wide-field microscope (scale bar = 100μm). EA, eye-antennal imaginal disc. OB, optic lobe.

Given the technical challenges to obtain such transcriptomic datasets at this point, we sought an alternative approach to test the point raised by reviewers. Our goal was to ask whether a differentiated cell, once mutated and tumorigenic, would return to a less differentiated state. To this end, we attempted to generate mutant clones at a later stage of eye-antennal imaginal development, when differentiation cues are already established, so that we could assess whether these late mutant cells would still lack differentiation markers.

First, we defined a time-point when the wave of differentiation has started and cells received signals to differentiation. This wave of differentiation follows the morphogenetic furrow (MF) and progresses in a posterior-to-anterior direction. In our hands, and as previously reported (Chanut and Heberlein, 1995), at early third instar (96h AEL) this differentiation wave has started. We could confirm this by the obvious presence of the MF (marked with white arrows, Author response image 2) and the ELAV positive staining of the posterior cells located further away from MF. However, there is some variation of the number of cells that express ELAV between different larvae, as you can appreciate between left and central pictures (Author response image 2). Additionally, using a later time point when there is less variability and more cells express ELAV (120h AEL) might not be ideal since discs would need to be dissected a day later, which would not allow sufficient time to robustly generate tumor clones.

**Author response image 2. respfig2:** Expression of ELAV (magenta) in developing eye-antennal imaginal discs (EA), 96h and 120h after egg laying (AEL). White arrows, morphogenetic furrow. DAPI, cyan; magenta, ELAV staining. Scale bar 100μm.

We thus tried to generate tumor clones at early 3^rd^ instar (96h AEL) where differentiation has started by using a heat shock flippase and later check the expression of ELAV (2 days after heat shock). In particular, we checked the ELAV expression in clones that are located in the most posterior region of the disc, making sure that these clones are generated after the MF was already anterior to them.

Our results showed that some clones generated in the posterior region of the disc (at 96h AEL) do not express ELAV (Author response image 3), suggestive of a dedifferentiation event in *ph*^*505*^-tumor clones. However, it is difficult to find clones in this region and in addition be entirely certain that this is indeed due to the tumor development and not simply a result of a developmental delay when clones were generated. Since we have generally used the eyFlp/MARCM system, we cannot completely exclude a potential effect of a developmental delay; hence, we tend to agree with the reviewers that one needs to be cautious in using the term dedifferentiation. In light of these considerations, and even though we observed some cases supporting our initial claim, we decided to re-word the manuscript to reflect a more accurate description of the condition (e.g. ‘failure to differentiate’).

**Author response image 3. respfig3:** Expression of ELAV in mitotic clones generated 96h AEL. Heat shock (HS) experiments were performed at 96h AEL (90min, 37ºC). Control experiments were done with FRT19A neutral clones (left side, GFP clones), where clones are very small at 6 days AEL, and with normal expression of ELAV. Mutant clones (GFP clones, central panel) were generated and some EA showed defects in ELAV patterning, as depicted in higher magnification in the panel on the right (white dashed arrow). Scale bar 100μm.

2) The control of the RNA-seq are the GFP-negative cells from FACS sorting. What about differential expression compared to wild type EA discs?

This project was designed with the aim of defining the transcriptional signature of *ph*^*505*^ tumors by comparing mutant cells with adjacent non-mutant cells. From the initial design, we found it important to define a tumor-signature that would be targetable by genetic tools specifically inside the mutant-clones. These considerations led us to selectively isolate GFP-positive (tumor cells) and GFP-negative cells which had undergone the exact same experimental procedure (including FACS sorting). One of the main benefits of this approach was that it allowed us to identify a mutant-specific signature and not a tumor tissue signature (mix of mutant and non-mutant cells), while minimizing artifacts generated by alternative experimental conditions. We believe this actually brings an advantage to the field regarding other published approaches using bulk material. The expression profiles of these GFP-negative cells also resembled the profiles of other published EA transcriptomes, which is also apparent from the clustering analysis based on TF expression levels (Figure 1 in the manuscript).

Nevertheless, in order to address the question posed by the reviewer, we performed differential expression analysis comparing our tumor transcriptomes against several available datasets of wild-type EA discs (which had already been included in our clustering analysis). The conclusions one can draw from such an approach are limited, as differences in gene expression levels through this analysis can arise due to batch-effects. Because of batch effects and the higher number of samples, we expected to find a higher number of differentially expressed genes, which was indeed the case. The total number of differentially expressed (DE) genes increased over 4-fold (padj<=0.01) when comparing GFP+/tumor with wild-type EA (Author response image 4). However, the general tendencies were conserved in both analyses; e.g. a higher number of genes is downregulated (DOWN vs. EA, Author response image 4) compared to up-regulated. Also, log2-fold-changes were correlated (r=0.624) between the two approaches.

**Author response image 4. respfig4:** Differential Expression analysis – comparison of *ph*^*505*^-tumor cells (GFP+) with wildtype EA discs. (**A**) Venn diagram showing the overlap (1088 DE genes) between the previous analysis (DE vs. GFP-) and the new analysis (DE vs. EA). (**B**) UpSet intersection diagram shows detail information about overlaps of down- and upregulated genes across two approaches of DE analysis. Some genes are exclusively found in some approaches (as 2764 genes are DOWN-reg in a GFP+ tumor vs. wt EA), while others are commonly found across analysis (as e.g. 859 DOWN- in both differential expression analysis). Common genes found across different approaches are depicted with connection line between the 2 fractions.

We found that from the 1337 DE genes obtained with our initial analysis (GFP+ vs. GFP-), 1088 (81%) are commonly found DE by comparing *ph*^*505*^-cells (GFP+) with wild-type EA discs. This included genes like *kni, cad* and *drm* that we tested as candidates in our RNAi screen. Moreover, we showed that knocking-down either of these 3 genes led to major reductions in tumor volume, revealing for the first time their importance in *ph*^*505*^-tumors.

We also note that our initial approach retrieved a considerable number of other genes (not detected in comparison to wt discs), supporting the power of our early approach. The comparison between GFP+ and GFP- cell populations recovered 249 DE genes not found when comparing to wild type tissues, 191 down- and 58 upregulated (UpSet plot, Author response image 4). Among these genes are e.g. Abd-A and *bgcn* (a germline differentiation factor that led to a 32% reduction in tumor volume when KD in *ph*^*505*^, similar to the effect observed when KD TFs as cad or drm). This example highlights the importance of comparing GFP+ and GFP- cells from the same tissues, as such targets would otherwise have been missed.

3) Among the 75 samples used to compare, why was Ras/Scrib not included? That is typically used and for this tumour plenty of microarray and RNA-seq data are available.

We have now included RNA-seq samples from RasV12/Scrib- eye-antennal tumors (Atkins et al., 2016) in our analysis (see modified Figure 1 in the manuscript), which were not available when this initial analysis was performed.

Including these samples did not alter the previous clustering results. Ph tumors cluster with early embryos while our control samples cluster with other EA disc samples (either wild-type or mutant). Thus, Ph tumors present a TF-signature that is different from other tumors (as RasV12/Scrib-) and more similar at this level to embryos.

4) How general are their findings? Do ph clones give rise to tumors in other tissues as well? The authors should test the effects of ph clones in other tissues, and address autonomy of ph clones effects.

We recognize the importance of assessing the effects of *ph* mutant clones in other tissues. We made significant efforts to address this point and expand the knowledge from previous observations. *ph* mutant clones, generated with a different allele (*ph*^*504*^), were previously shown to induce cell polarity defects with formation of cysts, and expression of Polycomb targets (e.g. Abd-B) in wing imaginal discs (Gandille et al., 2010). In addition, another study that used the cell-lethal system in combination with UbxFlp, showed that *ph*^*505*^ mutant cells lead to tissue overgrowth (Classen et al., 2009).

We also carried out some experiments ourselves, by assessing the effect of *ph*^*505*^ clones in wing discs and addressing the cell autonomy of these clones, as requested by the reviewers. We have generated *ph*^*505*^ clones using UbxFlp, and also observed cyst formation (Author response image 5’). These cysts, in analogy to what happened in EA discs, show ectopic expression of embryonic genes, as shown here for Caudal (Author response image 5’’-B’’’). This is suggestive of a cell fate switch, although at this stage we cannot exclude the possibility of the involvement of a slightly different transcriptional network, for which additional work would be needed.

**Author response image 5. respfig5:** Generation of *ph*^*505*^ mutant clones in wing imaginal discs. (**A**) Control clones generated with FRT19A blank and UbxFlp present notchy-shape (A’) and do not express Caudal (A’’). (**B**) *ph*^*505*^ clones generated in wing imaginal discs, similarly to what was observed in eye-antennal imaginal discs, have round shape (B’) and ectopically express the embryonic TF Caudal (Cad, B’’-B’’’). Scale bar 100μm.

As requested by the reviewers, we also assessed the non-autonomous effect in these tissues. We started by checking the activity of the JAK/STAT pathway, as it was previously shown to play a role in EA discs (Classen et al., 2009; Feng et al., 2011).

We found examples of wing discs that show a non-autonomous effect within the wing pouch (Author response image 6’-B’’’, white arrows), while the JAK/STAT reporter is not normally expressed in the pouch of wild-type wing discs (Author response image 6’).

In addition to this, and following on a previous observation, we also addressed the contribution of the JNK pathway to non-autonomous effects. For this purpose, we have checked the expression of a JNK target, MMP1 (metalloproteinase 1). Similarly to what we have shown in our manuscript for *ph*^*505*^ clones in EA discs, MMP1 is ectopically expressed in mutant clones, in comparison to control wing discs (Author response image 6’-D’). Moreover, in some clones MMP1 expression is observed in cells surrounding the clones (Author response image 6’’’), suggestive of a potential role for JNK in non-autonomy of these clones.

In addition to these observations in the wing disc, we have also examined other tissues within the laboratory. In summary, our unpublished observations revealed that *ph* mutant cells did not generate tumors in the brain (possibly due to cell death); Finally, some preliminary observations (which we aim to strengthen in the future) show that *ph* mutant cells can give rise to tumors in the adult fly gut (another epithelial tissue). Hence, there is a more general tumorigenic potential of *ph* mutations, which seems to be particularly impactful in epithelial tissues (which are the tissue type where many human tumors also arise, e.g. carcinomas).

**Author response image 6. respfig6:** JAK/STAT and JNK (MMP1) pathway activation in *ph^505^* clones in wing imaginal discs. (**A**) Expression of the 10x-STAT reporter in wild-type wing imaginal disc. (**B**) *ph*^*505*^ clones generated in wing imaginal discs show ectopic expression of the reporter inside the pouch (B’-B’’’). (**C**) MMP1 expression in wild-type wing imaginal disc. Ectopic expression of MMP1, a target of JNK, in *ph*^*505*^ clones (**D**). Inset (D’’’) showing non-autonomous expression of MMP1, suggestive of a role for JNK in non-autonomous effect of these clones. Scale bar 100μm.

5) Autonomous vs. non-autonomous effects should be addressed: ELAV that marks photoreceptors (PR) is used as a differentiation marker. ph clones lose ELAV and it is re-established when kni, the embryonic TF, is knocked down or Ato, required for PR differentiation, is expressed in these cells. What is puzzling is that adult eyes from discs with ph tumors display normal looking PRs in higher numbers. If ph mutant cells themselves are ELAV negative, then the surrounding wild-type cells must be forming these excessive PRs. As the authors cite (Feng et al., 2011), there is a non-autonomous component to the ph phenotype, which was proposed to be caused by JAK/STAT induction in neighbouring cells. So, are the tumors formed by the ph cells or their wt neighbours? This is a very important point as it changes the way the RNA-seq data would be interpreted. The authors state in the Discussion that the differential expression analysis was done comparing GFP (+) and (-) cells in discs with ph mutants. I think, comparing GFP (+) cells from discs with ph mutants to GFP (+) cells from discs with wt clones would be important as well, the same goes for GFP (-) cells.

The point that the reviewers raise is very valid, and we tried to tackle this aspect in several ways. First, we performed a quantitative analysis in order to assess the volume taken by ELAV+ (differentiated) cells in the eye disc (Author response image 7), and indeed we observed that there are more PR precursor cells in discs carrying tumors (shown in the blue bar plots on the left). As *ph*^*505*^-cells are mostly negative for ELAV, we focused on the surrounding non-mutant cells (GFP-) that might be responsible for formation of excessive number of normal looking PRs. We confirmed that the difference observed was attributed to the surrounding cells (purple bar plots on the right), in agreement with the reviewers’ point.

However, if these non-mutant cells were tumorigenic, one would expect that upon transplantation they would also contribute to the tumor mass, which we never observed, however. Hence, this strengthens our position that tumorigenic cells are GFP+ *ph* mutant cells, while the surrounding tissue is not (even if it may overgrow due to non-autonomous effects, as previously shown (Feng et al., 2011)).

**Author response image 7. respfig7:** Quantitative analysis of tissue volume occupied by ELAV+ cells. Number of tissues used for quantification analysis (using Imaris): FRT19A N=12; *ph*^*505*^ N=19). Trend of increase in absolute volume of ELAV+ inside tissues carrying *ph*^*505*^-tumor clones is observed, in comparison with FRT19A (in blue). This trend is enhanced and significantly different when looking specifically at volumes outside of clones and thus GFP- (in purple). Blue bars – total volume of ELAV+ *per* tissue. Purple bars – fraction of ELAV+ that does not overlap with GFP, and thus fraction ‘outside of clones’. * = p<0.05.

It is important to note that the phenotype that triggered the concern (‘higher number of PRs’) has a low penetrance, as only 20% of larvae carrying tumor tissues eclose (see Figure 1—figure supplement 1). It becomes difficult to disentangle this issue, as one cannot be certain about which larvae survive to adulthood (e.g. the surviving 20% could actually be among the ones that were composed of smaller tissues and smaller tumor clones). Nevertheless, we still tried to address this point in an unbiased and quantitative manner (see above), leading us to conclude that the overgrowth phenotype observed in the eclosing adults is likely due to compensatory mechanisms during the development of eye-antennal disc with formation of tumor clones.

Furthermore, observations made with the cell-lethal system in EA discs (Classen et al., 2009) showed that even in the absence of non-mutant surrounding cells, the tissue still overgrows. Thus, the overgrowth phenotype is not exclusively due to wt cells, but they do contribute and support this phenotype (see also our response to point 4).

Taken together, these observations indicate to us that it is important to distinguish between tumorigenic (mutant) and overgrowth phenotypes. Based on previous reasoning, we consider that further comparison of GFP+ and GFP- transcriptomes from wild-type tissues could be of interest to distinguish these two populations, but this approach would be technically challenging (wild-type tissues are 4-times smaller, see Figure 2—figure supplement 1), and would diverge from the focus of our work.

It was very challenging to set up the workflow for sample preparation for RNA-seq from *ph*^*505*^-tissues. We believe that the suggested experiment would require an even higher number of tissues (perhaps over 1000 discs), and their dissection would also need to be done in less than an hour to avoid cell death, making it impossible for us to achieve at this point.

In sum, the design of this study was targeted at comparing mutant cells with adjacent cells (non-mutant) within the same tissue. It is possible that adjacent cells contribute to compensatory growth (Uhlirova et al., 2005). One of the strong points from our work is the extended in vivo validation presented in the manuscript. This serves as good evidence that our strategy has been successful in identifying key regulators in mutant cells, and with a functional validation through KD which led to considerable effects in the tumorigenic capacity and overall tissue overgrowth.

6) Is Knirps a key regulator of the downregulated genes? The authors speculate on this point, but should include the analysis of the downregulated genes in iRegulon as this would reveal if Knirps is implicated as a key regulator in both sets.

The initial analysis using iRegulon (as mentioned in the manuscript) was done with all DE genes (1337) as input. Indeed, Knirps, among others, was enriched as a regulator of targets that are DE in our dataset. The question posed by the reviewers is one that we had already asked ourselves: Knirps is a well-known short-range repressor, leading us to consider if Knirps could be partially responsible for the putative repression of downregulated genes. We have now extended the iRegulon analysis, partitioned into downregulated genes as input. The output revealed that Knirps is indeed among the possible regulators of these genes, in agreement with its repressive function. Conversely, when we provide the list of upregulated genes as input, Knirps ectopic expression does not seem to be directly contributing (at least from motif enrichment analysis), which is expected given its main role as a repressor. We have also clarified the main text of the manuscript.

7) Ato overexpressing clones in normal tissue appear to be small. The ELAV pattern in these discs is also somewhat disrupted. While overexpression of Ato clearly prevents tumor overgrowth in this model, it is unclear if this is due to actually inducing differentiation or if Ato overexpression simply prevents cells from proliferating. The authors should clarify this point, which may require EdU or pH3 stainings to address the proliferation, along with quantifications of normal relative to tumor clones sizes.

We understand the concern raised by the reviewers. Indeed, it has been previously shown that atonal reduces ectopic proliferation in ‘eyeful discs’ (Bossuyt et al., 2009), highlighting the relevance of this question.

We went further to characterize if Ato overexpression prevents *ph*^*505*^-tumor cells from proliferating (see modified text in the manuscript and the addition of new data in Figure 6 and new Figure 6—figure supplement 2). We performed phospho-histone H3 (pH3) stainings in the four conditions of interest, followed by quantitative analysis (see Materials and methods). The results show that a higher number of proliferating cells was present in *ph^505^*-tumor tissues in comparison to control tissues (FRT19A). Moreover, this overall increase in proliferation was largely due to cells outside of *ph*^*505*^ clones (Figure 6—figure supplement 2). Forcing expression of atonal in *ph^505^* mutant cells resulted in an overall reduction of proliferation in these tissues, due to a reduced number of proliferating cells inside mutant clones co-expressing atonal (Figure 6 in the manuscript).

In summary, atonal expression prevents cells from proliferating and thus counterbalance the overgrowth phenotype of *ph*^*505*^ cells. The expression of atonal in *ph*^*505*^clones led to a dramatic reduction of tumor burden (from 46% to 3%), and ultimately to a normal eye differentiation pattern. This question was thus helpful for us to obtain detailed information, as highlighted in the modified manuscript.